# Mixture of Noise for Pre-Trained Model-Based Class-Incremental Learning

**Kai Jiang**[1,2], **Zhengyan Shi**[2,3], **Dell Zhang**[2], **Hongyuan Zhang**[2,4*] **and Xuelong Li**[2*]

[1]School of Artificial Intelligence, OPtics and ElectroNics, Northwestern Polytechnical University
[2]Institute of Artificial Intelligence (TeleAI) of China Telecom
[3]School of Computer Science & School of Artificial Intelligence, Shanghai Jiao Tong University
[4]University of Hong Kong
jk@mail.nwpu.edu.cn, macho2021@sjtu.edu.cn, dell.z@ieee.org,
hyzhang98@gmail.com, xuelong_li@ieee.org

## Abstract

Class Incremental Learning (CIL) aims to continuously learn new categories while retaining the knowledge of old ones. Pre-trained models (PTMs) show promising capabilities in CIL. However, existing approaches that apply lightweight fine-tuning to backbones still induce parameter drift, thereby compromising the generalization capability of pre-trained models. Parameter drift can be conceptualized as a form of noise that obscures critical patterns learned for previous tasks. However, recent researches have shown that noise is not always harmful. For example, the large number of visual patterns learned from pre-training can be easily abused by a single task, and introducing appropriate noise can suppress some low-correlation features, thus leaving a margin for future tasks. To this end, we propose learning beneficial noise for CIL guided by information theory and propose **Mi**xture of **N**oise (MIN), aiming to mitigate the degradation of backbone generalization from adapting new tasks. Specifically, task-specific noise is learned from high-dimension features of new tasks. Then, a set of weights is adjusted dynamically for optimal mixture of different task noise. Finally, MIN embeds the beneficial noise into the intermediate features to mask the response of inefficient patterns. Extensive experiments on six benchmark datasets demonstrate that MIN achieves state-of-the-art performance in most incremental settings, with particularly outstanding results in 50-steps incremental settings. This shows the significant potential for beneficial noise in continual learning. Code is available at `https://github.com/ASCIIJK/MiN-NeurIPS2025`.

## 1 Introduction

Unlike human beings can naturally learn new concepts without forgetting, existing AI systems lack the ability for continual learning [1–4]. To address this challenge, Class-Incremental Learning (CIL) is proposed to mitigate catastrophic forgetting [5] of old knowledge when acquiring new concepts. Most traditional CIL works focus on learning-from-scratch [6–12], i.e., learning a model without pre-training. However, the pre-training technique has been widely used in real-world applications [13, 14]. Fine-tuning with pre-trained model (PTM) on downstream tasks has become a consensus within the AI community. This has sparked significant research interest in the continual fine-tuning of PTMs for downstream tasks.

Due to pre-training on large-scale datasets, PTMs appear strong generalization. Therefore, preserving the generalization ability of PTMs across all incremental phases becomes a key factor to maintain

---

*Corresponding authors

39th Conference on Neural Information Processing Systems (NeurIPS 2025).

the performance in downstream tasks. A critical challenge arises from persistent cross-task feature interference. Pre-trained models inherently exhibit task-specific redundancy, with only limited discriminative features being task-relevant. During continual adaptation, models inadvertently assimilate non-essential features into the decision boundaries of the current task. These features induce catastrophic forgetting through two mechanisms: 1) when the features are inherited from previous tasks, they compromise established decision boundaries of preceding models, 2) adoption of these features by subsequent tasks disrupts current task boundaries. This issue stems from the parameter sensitivity of PTMs. While such models demonstrate remarkable adaptability to new tasks, this flexibility conversely renders them vulnerable to interference from subsequent tasks. Rather than pursuing enhanced feature utilization efficiency during task adaptation, it is advisable to mitigate individual tasks' dependence on redundant features.

Positive-incentive noise (Pi-Noise) has been proven effective in diverse tasks [1]. It works by masking the confusing part among different categories with noise and highlighting the recognizable part. Catastrophic forgetting stems from parameter drift for old tasks, it can be conceptualized as the intrusion of noise during learning a new task, which undermining the original pattern of the old framework. Joint training based on old tasks also changes the original parameters, but the noise introduced by joint training suppresses confusing inter-task patterns, consequently producing a positive impact. Therefore, we aim to **directly learn this kind of positive noise from new tasks** and embed it into intermediate features. The main contributions of our work are three-fold:

- We rethink class incremental learning from the perspective of noise, interpreting parameter drift as introducing noise that is destructive to old tasks. Based on this assumption, we propose to learn beneficial noise for new tasks to suppress the response of the confusing pattern among tasks.

- We design the *Noise Expansion* strategy to learn a new noise generation module for each task. To avoid multiple inference through the backbone, we mix the beneficial noise from different tasks by training a set of learnable weights using an auxiliary classifier and residual loss, dubbed the *Noise Mixture* strategy. The noise mixture from multiple tasks is embedded with intermediate features to suppress the confusing patterns among tasks.

- We validate MIN on six widely used benchmark datasets. Experimental results show that MIN only needs few learnable parameters to learn beneficial noise adapted to the new task and achieve state-of-the-art performance. Additionally, the visualization with Grad-CAM indicates that MIN effectively suppresses the response of the irrelevant region and strengthens the representation of key pattern.

## 2 Related Work

**Class-Incremental Learning (CIL)**: aims to learn a unified model from a data stream without forgetting. Conventional CIL approaches can be roughly divided into several types. Rehearsal-based methods [8, 15–18] construct a fixed exemplar-set to preserve a fraction of samples from old tasks for future training. Regularization-based methods add parameter regularization terms [6, 7] or function regularization terms [9, 10, 19, 20] to transfer knowledge from the old model to the new one. Architecture-based methods [11, 21–24, 12, 25] often add new trainable modules to expand the feature space for new tasks. Analytic learning-based methods [26–29] reformulate the CIL procedure into a recursive analytic learning process.

**Pre-Trained Model-Based CIL**: aims to fine-tune the PTMs in sequential downstream tasks without forgetting while keeping the generalization ability of PTMs. L2P [30] and DualPrompt [31] construct a prompt pool to select the suitable prompts for samples during model inference. CODA-Prompt [32] develops the prompt selection process with attention-based weighting. SLCA [33] slows the update of the backbone and designs a classifier alignment strategy using Gaussian modeling for previous classes. APER [34] shows that the prototypical classifier is a strong baseline. FeCAM [35] explores the prototype network and shows that modeling the feature covariance relations is better than previous attempts at sampling features from normal distributions and training a linear classifier. RanPAC [36] adopts random projection to enhance the linear separability for prototype-based CIL. EASE [37] adds task-specific branches for feature expansion. COFiMA [38] selectively weighs each parameter in the weights ensemble by leveraging the Fisher information of the weights of the model. MOS [39] rectifies the model with a training-free self-refined adapter retrieval mechanism during inference.

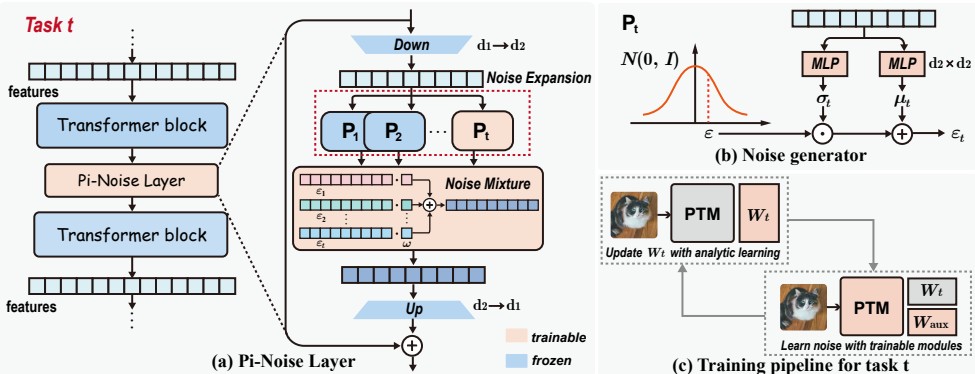

Figure 1: Illustration of MIN. **(a) Pi-Noise Layer:** is inserted between transformer blocks of PTM for learning positive noise. *Noise Expansion* adds a generator for each task and *Noise Mixture* learns to combine positive noise from different tasks. **(b) Noise generator:** samples a signal from a standard normal distribution and then transfers it into noise using the mean and variance vectors generated by MLP layers. **(c) Training pipeline for task** $t$**:** consists of 3 steps. It firstly updates classifier with analytic learning and then learns noise with trainable modules. Finally, it updates classifier again with analytic learning.

**Positive-incentive Noise (Pi-Noise)**: is the random signal that can simplify the target task which is ubiquitous in diverse fields [1, 40–42]. Following the framework of Pi-Noise, [43] proposes an approximate method to learn Pi-noise via variational inference, which validates the efficacy of the idea about generating random noise to enhance the classifier. [44] designs an auxiliary Gaussian distribution related to contrastive loss to define task entropy and find that the developed Pi-Noise generator successfully learns augmentations on vision datasets. [45] adopts Pi-Noise to design a fine-tuning method towards vision-language alignment. In addition, [46] apply the Pi-Noise to graph contrastive learning, thus adding edges adaptively with low time and memory burden.

## 3 Preliminary

**Class Incremental Learning**: learns a unified model from a data stream. Assume that the data steam $\mathcal{D} = \{\mathcal{D}_1, \mathcal{D}_2 \cdots, \mathcal{D}_T\}$ consists of $T$ tasks, with each task containing several disjoint categories. For task $t$, $\mathcal{D}_t = \{(x_i^t, y_i^t)\}_{i=1}^{n_t}$ denotes the current training set with $n_t$ instances. $x_i^t \in \mathcal{X}_t$ is the input image and $y_i^t \in \mathcal{Y}_t$ is the corresponding label, where $\mathcal{X}_t$ and $\mathcal{Y}_t$ denote the image set and the label set, respectively. Specially, we denote the label set of a new task and old tasks as $\mathcal{Y}_{\text{new}} = \mathcal{Y}_t$ and $\mathcal{Y}_{\text{old}} = \mathcal{Y}_1 \cup \mathcal{Y}_2 \cup \cdots \cup \mathcal{Y}_{t-1}$, respectively, with $\mathcal{Y}_{\text{new}} \cap \mathcal{Y}_{\text{old}} = \emptyset$. $|\mathcal{Y}_{\text{new}}| = K$ is denoted as the number of new categories, while $|\mathcal{Y}_{\text{old}}| = M$ is denoted as the number of old categories. We follow the exemplar-free setting [30, 39], where no data from previous tasks is retained in subsequent tasks. Only $\mathcal{D}_t$ can be accessed in task $t$. After learning each task, the model will be tested on all seen categories. By leveraging the PTM for initialization, we decompose the model $\mathbb{F}_t = \{\mathcal{F}_t, W_t\}$ into two distinct components, i.e., the pre-trained backbone $\mathcal{F}_t$ and classifier $W_t$. Following typical PTM-based CIL works [30, 39], the most parameters of pre-trained backbone are frozen except a little part for fine-tuning.

**Analytic Learning**: develops a new branch for class-incremental learning [26, 27]. It iteratively solves the classifier weights through linear regression. For each task, only the current task data, the classifier weights from the previous task, and a fixed-size autocorrelation matrix are required to determine the new task classifier weights. Specifically, the classification problem can be defined in the form of linear regression, as shown in Eq. (1),

$$\underset{W}{\arg\min} \|\mathcal{Y} - \mathcal{F}(\mathcal{X}) W\|_2^2 + \lambda \|W\|_2^2, \tag{1}$$

where $\mathcal{X}$ denotes the image set, $\mathcal{Y}$ the corresponding label set, $\mathcal{F}$ the feature extractor and $W$ the classifier weight to be solved. The optimal solution to Eq. (1) can be found in

$$W = \left(\mathcal{F}(\mathcal{X})^{\mathsf{T}} \mathcal{F}(\mathcal{X}) + \lambda I\right)^{-1} \mathcal{F}(\mathcal{X})^{\mathsf{T}} \mathcal{Y}. \tag{2}$$

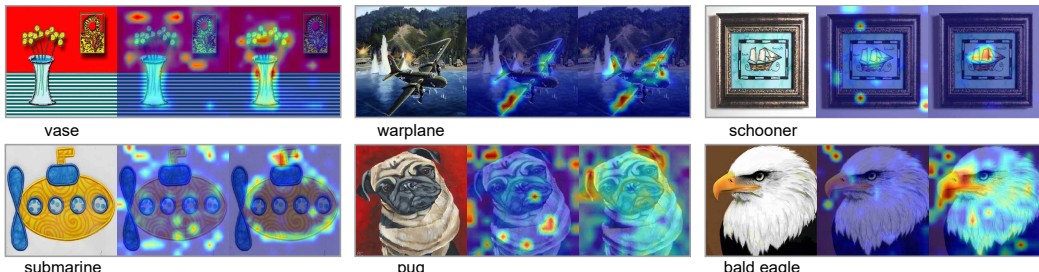

Figure 2: Grad-CAM visualization of the baseline method and MIN. The first image of each group is the original image, the second is the visualization of the baseline method, and the third image is the visualization result of the MIN. In addition, the first row shows the visualization where both methods achieve correct classification. In the second row, the baseline method is misclassified, while the MIN maintains correct.

According to [26], Eq. (2) can be rewritten in the iterative form shown in Eq. (3) for solving the classifier weight,

$$W_t = \left[ W_{t-1} - R_t \mathcal{F}(\mathcal{X}_t)^\mathsf{T} \mathcal{F}(\mathcal{X}_t) W_{t-1}, R_t \mathcal{F}(\mathcal{X}_t)^\mathsf{T} \mathcal{Y}_t \right], \tag{3}$$

where $W_{t-1}$ denotes the classifier weight at task $t-1$, $\mathcal{X}_t$ denotes the image set at task $t$, $\mathcal{Y}_t$ the corresponding label set. $R_t$ is the autocorrelation matrix, which is found according to Eq. (4),

$$\begin{aligned} B_t &= \left( I + \mathcal{F}(\mathcal{X}_t) R_{t-1} \mathcal{F}(\mathcal{X}_t)^\mathsf{T} \right)^{-1}, \\ R_t &= R_{t-1} - R_{t-1} \mathcal{F}(\mathcal{X}_t)^\mathsf{T} B_t \mathcal{F}(\mathcal{X}_t) R_{t-1}. \end{aligned} \tag{4}$$

This method establishes a strong baseline [26], and the classifier derived from it serves as the foundation of our work.

## 4 The proposed approach: MIN

Although PTMs exhibit strong generalization capabilities, continuously fine-tuning in downstream tasks leads to rapid performance degradation. The degradation of generalization capability stems from parameter drift, which introduced a destructive noise to obscure the recognition pattern of the previous task. Therefore, we aim to **learn positive noise to adapt a new task while freezing the original parameters**. Continually learning tasks in sequence results in multiple noises from different tasks during inference. To avoid multiple inference through the backbone, we need to **integrate these noises from different tasks within each module**. In this section, we first introduce *Noise Expansion* to add a trainable noise learner for each task. Then, we adopt *Noise Mixture* to integrate these noises and embed the mixture of noise to intermediate features. As illustrated in Fig. 2, the visualization shows the impact of the proposed methods for subsequent tasks.

### 4.1 Noise Expansion

Due to the absence of guidance from the information among tasks, fine-tuning PTM is prone to absorb some irrelevant features to determine decision boundaries. It is equivalent to introducing noise for the current task. Since the label information within the task is available, this noise cannot affect the current task. But it may constitute the decision boundary of previous tasks and thus result in parameter drift to affect the classification pattern of old categories. To this end, we model the parameter drift as a kind of noise. Following [1], **noise is not always harmful**. For example, the noise introduced by joint training can play a positive role in classification. Therefore, we aim to **learn positive noise to suppress the confusing pattern across tasks** through a large interference to mask irrelevant activation in the features.

Suppose that the pre-trained backbone $\mathcal{F} = \{f_1, \cdots, f_L\}$ consists of $L$ blocks. Its forward propagation process can be represented by Eq. (5) and (6),

$$r_1 = f_1(x), \tag{5}$$

$$r_l = f_l(r_{l-1}), \tag{6}$$

where $x$ denotes the input images and $f_l$ the $l^{th}$ block. As illustrated in Fig. 1(a), we design Pi-Noise layers between these blocks, thereby modifying the forward inference process of the backbone network as Eq. (7),

$$r_l = \mathcal{P}_l \left( f_l \left( r_{l-1} \right) \right), \tag{7}$$

where $\mathcal{P}_l$ denotes the $l^{th}$ Pi-Noise layer. Due to the absence of inter-task information guidance, employing a single Pi-Noise proves inadequate for effectively simplifying all tasks, thereby causing beneficial noise to degenerate into negative noise. To address this limitation, we suggest learning distinct noise for each task, with the parameters in these task-specific generators remaining mutually independent.

To improve efficiency of parameter utilization, we first reduce the dimensionality of the input features to minimize the scale of each noise generator. Specifically, we employ a down-projection layer $W_{\text{down}} \in \mathbb{R}^{d_1 \times d_2}$ to reduce the feature dimension from $d_1$ to $d_2$. After generating the noise, we use an up-projection layer $W_{\text{up}} \in \mathbb{R}^{d_2 \times d_1}$ to remap the dimension of noise back to $d_1$. Notably, both the down-projection layer $W_{\text{down}}$ and the up-projection layer $W_{\text{up}}$ are shared across all tasks, with **each of their elements sampled from a standard normal distribution**. For task $t$, $P_t$ is employed to generate the corresponding beneficial noise, which is composed of two MLP layers denoted

---

**Algorithm 1:** Training pipeline for MIN

**Input:** Incremental Datasets: $\{\mathcal{D}_1, \cdots, \mathcal{D}_T\}$,
      Pre-trained backbone:
      $\mathcal{F} = \{f_1, \cdots, f_L\}$
**Output:** Incrementally trained model

1 **for** $t = 1$ **to** $T$ **do**
2      Update the classifier $W_t$ via Eq. (3)
3      Expand the new noise generator $P_t$
4      Initialize the weight $\omega$ via Eq. (12)
5      Construct an auxiliary classifier $W_{\text{aux}}$
6      Optimize the $\omega$, $P_t$ and $W_{\text{aux}}$ via Eq. (14)
7      Update the $W_t$ via Eq. (3)
8      Test the model
9 **end**
10 **return** the updated model

---

as $\phi_t^\mu$ and $\phi_t^\sigma$, responsible for producing the mean vector $\mu_t$ and variance vector $\sigma_t$, respectively. We sample a random signal $\varepsilon$ from a normal distribution and transform it into beneficial noise as described in Eq. (8). Using this reparameterization technique, we can train $P_t$ via backpropagation.

$$\varepsilon_t = \varepsilon \cdot \sigma_t + \mu_t. \tag{8}$$

In summary, the process of deriving beneficial noise from the input feature $r_l$ is illustrated in Eq. (9):

$$\varepsilon_t = \varepsilon \cdot \phi_t^\sigma \left( r_l W_{\text{down}} \right) + \phi_t^\mu \left( r_l W_{\text{down}} \right). \tag{9}$$

For an intermediate feature $r_l$, we obtain a set of noise denoted as $\{\varepsilon_1, \cdots, \varepsilon_t\}$. By combining these noise instances, we generate a noise mixture, which is then mapped back to the $d_1$-dimensional space via the up-projection layer $W_{\text{up}}$ to adjust the intermediate feature $r_l$,

$$\hat{r}_l = r_l + \varphi \left( \{\varepsilon_1, \cdots, \varepsilon_t\} \right) W_{\text{up}}, \tag{10}$$

where $\varphi \left( \cdot \right)$ denotes the noise mixing operation, which is detailed in the next section. For the current task, only the noise generator $P_t$ is trained, while all noise generators remain frozen in subsequent tasks. The structure of $P_t$ is simple, containing only two MLP layers. Since $d_2 \ll d_1$, the number of parameters trained for each task is approximately equivalent to two $d_2 \times d_2$ square matrices.

### 4.2 Noise Mixture

Through the *Noise Expansion*, we obtain a set of noise denoted as $\{\varepsilon_1, \ldots, \varepsilon_t\}$. If each noise is used separately for inference, it causes the test complexity to grow linearly with tasks like [37] and [39]. The simplest approach is to compute their average. However, substantial variations exist among different tasks. For task $t$, it may benefit from certain similar previous tasks, and thus the beneficial noise corresponding to those tasks should be assigned higher weights. Conversely, some tasks might conflict with the current task, making the noise derived from them detrimental to the current task. Therefore, when learning a new task, **it is critical to holistically consider the complexity of all tasks, ensuring that the noise mixture achieves an optimal trade-off between old and new tasksduring task adaptation**. To this end, we adjust all noise instances using a set of learnable weights $\omega$. First,

Table 1: Average and last performance comparison on CIFAR-100 and CUB-200 datasets with **ViT-B/16-IN21K** as the backbone. We report all compared methods with their source code. The best performance is shown in bold, and the second-best result is underlined. All methods are implemented without using exemplars.

| Methods | CIFAR-100 | | | | | | CUB-200 | | | | | |
| --- | --- | --- | --- | --- | --- | --- | --- | --- | --- | --- | --- | --- |
| | 10 steps | | 20 steps | | 50 steps | | 10 steps | | 20 steps | | 50 steps | |
| | $\overline{A}$ | $A_T$ | $\overline{A}$ | $A_T$ | $\overline{A}$ | $A_T$ | $\overline{A}$ | $A_T$ | $\overline{A}$ | $A_T$ | $\overline{A}$ | $A_T$ |
| L2P | $85.92_{\pm0.78}$ | $79.19_{\pm0.57}$ | $81.90_{\pm0.54}$ | $73.93_{\pm0.33}$ | $74.29_{\pm0.41}$ | $61.83_{\pm0.26}$ | $84.29_{\pm0.32}$ | $74.51_{\pm0.36}$ | $81.75_{\pm0.18}$ | $70.31_{\pm0.25}$ | $78.51_{\pm0.47}$ | $64.59_{\pm0.32}$ |
| DualPrompt | $89.65_{\pm0.55}$ | $84.89_{\pm0.79}$ | $85.57_{\pm0.62}$ | $79.27_{\pm0.85}$ | $73.66_{\pm0.35}$ | $63.97_{\pm0.41}$ | $84.39_{\pm0.54}$ | $73.45_{\pm0.75}$ | $83.79_{\pm0.65}$ | $72.48_{\pm1.05}$ | $78.06_{\pm0.45}$ | $64.80_{\pm0.82}$ |
| CODA-Prompt | $91.05_{\pm0.32}$ | $86.44_{\pm0.87}$ | $87.51_{\pm0.70}$ | $80.87_{\pm0.25}$ | $69.54_{\pm0.58}$ | $55.66_{\pm0.48}$ | $84.15_{\pm0.87}$ | $74.00_{\pm1.05}$ | $83.89_{\pm0.37}$ | $71.33_{\pm0.55}$ | $75.66_{\pm0.25}$ | $61.83_{\pm0.32}$ |
| ACIL | $91.21_{\pm0.03}$ | $86.74_{\pm0.05}$ | $91.34_{\pm0.03}$ | $86.78_{\pm0.02}$ | $91.51_{\pm0.07}$ | $86.84_{\pm0.15}$ | $91.74_{\pm0.21}$ | $87.22_{\pm0.16}$ | $91.83_{\pm0.18}$ | $87.09_{\pm0.17}$ | $91.73_{\pm0.33}$ | $86.87_{\pm0.19}$ |
| SLCA | $92.67_{\pm0.89}$ | $89.30_{\pm0.44}$ | $93.32_{\pm0.76}$ | $88.21_{\pm0.55}$ | $90.76_{\pm0.30}$ | $84.52_{\pm0.57}$ | $86.83_{\pm0.44}$ | $79.47_{\pm0.57}$ | $83.38_{\pm0.34}$ | $74.77_{\pm0.28}$ | $76.60_{\pm0.47}$ | $62.51_{\pm0.62}$ |
| FeCAM | $93.23_{\pm0.15}$ | $89.05_{\pm0.17}$ | $91.86_{\pm0.11}$ | $87.04_{\pm0.26}$ | $90.92_{\pm0.17}$ | $85.34_{\pm0.39}$ | $92.73_{\pm0.27}$ | $88.38_{\pm0.33}$ | $92.89_{\pm0.25}$ | $88.42_{\pm0.20}$ | $91.16_{\pm0.34}$ | $85.88_{\pm0.43}$ |
| RanPAC | $93.25_{\pm0.07}$ | $89.55_{\pm0.11}$ | $91.81_{\pm0.15}$ | $88.69_{\pm0.12}$ | $91.75_{\pm0.10}$ | $87.19_{\pm0.08}$ | $\underline{93.30}_{\pm0.14}$ | $\underline{89.78}_{\pm0.15}$ | $\underline{93.30}_{\pm0.17}$ | $89.27_{\pm0.20}$ | – | – |
| APER | $92.22_{\pm0.05}$ | $87.45_{\pm0.14}$ | $90.57_{\pm0.09}$ | $85.03_{\pm0.11}$ | $88.75_{\pm0.09}$ | $82.37_{\pm0.08}$ | $88.32_{\pm0.15}$ | $85.75_{\pm0.10}$ | $88.34_{\pm0.08}$ | $85.42_{\pm0.05}$ | $88.42_{\pm0.13}$ | $85.29_{\pm0.15}$ |
| EASE | $92.11_{\pm0.27}$ | $87.72_{\pm0.88}$ | $91.51_{\pm0.33}$ | $85.80_{\pm0.52}$ | $84.31_{\pm0.25}$ | $74.47_{\pm0.68}$ | $88.17_{\pm0.35}$ | $79.64_{\pm0.88}$ | $83.52_{\pm1.22}$ | $74.77_{\pm1.05}$ | $91.54_{\pm0.44}$ | $86.51_{\pm0.32}$ |
| COFiMA | $\underline{93.87}_{\pm0.16}$ | $89.77_{\pm0.08}$ | $92.86_{\pm0.27}$ | $88.09_{\pm0.11}$ | – | – | $88.17_{\pm0.35}$ | $79.64_{\pm0.88}$ | $83.52_{\pm1.22}$ | $74.77_{\pm1.05}$ | – | – |
| MOS | $93.83_{\pm0.12}$ | $\underline{90.19}_{\pm0.21}$ | $\underline{93.10}_{\pm0.06}$ | $\underline{89.10}_{\pm0.25}$ | $\underline{92.36}_{\pm0.04}$ | $\underline{87.39}_{\pm0.18}$ | $92.08_{\pm0.30}$ | $88.17_{\pm0.46}$ | $92.62_{\pm0.35}$ | $\underline{88.51}_{\pm0.18}$ | $\underline{91.99}_{\pm0.12}$ | $\underline{87.59}_{\pm0.20}$ |
| **MIN (Ours)** | $\mathbf{95.12}_{\pm0.05}$ | $\mathbf{92.12}_{\pm0.16}$ | $\mathbf{94.31}_{\pm0.04}$ | $\mathbf{91.03}_{\pm0.29}$ | $\mathbf{93.63}_{\pm0.20}$ | $\mathbf{89.82}_{\pm0.36}$ | $\mathbf{94.00}_{\pm0.15}$ | $\mathbf{91.22}_{\pm0.18}$ | $\mathbf{93.84}_{\pm0.12}$ | $\mathbf{90.54}_{\pm0.14}$ | $\mathbf{93.21}_{\pm0.14}$ | $\mathbf{89.95}_{\pm0.22}$ |

initialize the $\omega$ as Eq. 12.

$$s_{t,i} = \frac{\mu_t \cdot \mu_i}{\|\mu_t\| \, \|\mu_i\|}, \tag{11}$$

$$w_i = \frac{\exp\left(s_{t,i}/\tau\right)}{\sum_{j=1}^{t} \exp\left(s_{t,j}/\tau\right)}, \tag{12}$$

where $\mu_i$ denotes the task prototype saved from the task $i$, $\tau$ is the temperature coefficient and is set to 2 by default. The noise mixing operation is expressed as Eq. (13),

$$\varphi\left(\{\varepsilon_1, \cdots, \varepsilon_t\}\right) = \sum_{i=1}^{t} \varepsilon_i \omega_i. \tag{13}$$

Then, the output feature $r_L$ of the backbone is given by Eq. (5) and (7).

$$\mathcal{L}_{cls} = \ell\left(z_L W_{\text{aux}}, y - z_L W_t\right). \tag{14}$$

For the training pipeline, we first update the classifier $W_t$ according to Eq. (3) and construct an auxiliary classifier $W_{\text{aux}}$ with all elements initialized to zero. Then, we update $P_t$ and $W_{\text{aux}}$ using the cross-entropy loss $\ell\left(\cdot\right)$ to fit the residual between the output logits $z_L W_t$ and the ground truth according to Eq. (14). It worth noting that the classifier $W_t$ does not participate in gradient updates during training. After training, we update $W_t$ once more with the updated $P_t$ according to Eq. (3) while $W_{\text{aux}}$ is discarded. We summarize the training pipeline of MIN in Algorithm 1.

## 5 Experiments

In this section, we evaluate MIN on several benchmark datasets and compare it with other SOTA methods to demonstrate its superiority. In addition, we provide an ablation study and a visualized analysis to validate the effectiveness of MIN. More experimental results can be found in the supplementary.

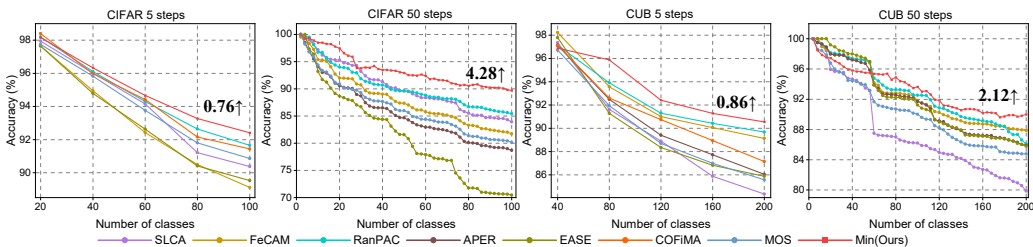

Figure 3: The performance of each learning session under different settings of CIFAR-100 and CUB-200. All methods are initialized with **ViT-B/16-IN1K**.

Table 2: Average and last performance comparison on ImageNet-A and ImageNet-R datasets with **ViT-B/16-IN21K** as the backbone. We report all compared methods with their source code. The best performance is shown in bold, and the second-best result is underlined. All methods are implemented without using exemplars.

| Methods | ImageNet-A | | | | | | ImageNet-R | | | | | |
|---|---|---|---|---|---|---|---|---|---|---|---|---|
| | 10 steps | | 20 steps | | 50 steps | | 10 steps | | 20 steps | | 50 steps | |
| | $\overline{A}$ | $A_T$ | $\overline{A}$ | $A_T$ | $\overline{A}$ | $A_T$ | $\overline{A}$ | $A_T$ | $\overline{A}$ | $A_T$ | $\overline{A}$ | $A_T$ |
| L2P | 54.90±1.22 | 43.12±1.06 | 54.25±1.35 | 41.41±2.08 | 49.89±0.88 | 36.41±0.62 | 66.66±0.36 | 59.78±0.56 | 63.75±0.36 | 55.78±0.45 | 57.86±0.22 | 48.33±0.18 |
| DualPrompt | 52.89±0.85 | 39.89±0.65 | 50.36±1.02 | 36.87±0.84 | 43.85±0.77 | 29.95±1.06 | 70.29±0.25 | 66.50±0.65 | 66.52±0.33 | 61.77±0.40 | 55.38±0.28 | 47.40±0.22 |
| CODA-Prompt | 50.16±1.35 | 40.42±1.27 | 49.19±2.08 | 38.05±1.55 | 38.24±0.66 | 26.60±0.82 | 76.76±0.59 | 72.98±0.47 | 70.45±0.56 | 64.68±0.51 | 55.82±0.36 | 45.43±0.12 |
| ACIL | 63.30±0.42 | 53.26±0.66 | 62.45±0.57 | 52.28±0.55 | 62.48±0.36 | 50.92±0.21 | 75.17±0.05 | 68.40±0.22 | 75.42±0.08 | 68.19±0.16 | 75.37±0.09 | 67.82±0.16 |
| SLCA | – | – | – | – | – | – | 82.26±0.26 | 76.82±0.34 | 81.85±0.22 | 76.63±0.28 | 79.35±0.31 | 72.87±0.18 |
| FeCAM | 56.04±1.05 | 46.41±0.65 | 55.41±1.15 | 45.29±0.94 | 55.36±0.85 | 45.16±0.65 | 79.02±0.22 | 72.53±0.18 | 77.84±0.15 | 71.05±0.26 | 63.69±0.17 | 55.93±0.11 |
| RanPAC | 64.61±1.35 | 54.05±0.45 | 62.37±1.07 | 46.61±0.65 | – | – | 82.12±0.31 | 77.55±0.16 | 77.88±0.09 | 72.27±0.15 | 75.62±0.27 | 69.28±0.16 |
| APER | 60.26±0.72 | 48.91±0.66 | 60.84±1.05 | 48.78±0.36 | 61.29±0.82 | 48.58±0.26 | 75.45±0.11 | 67.32±0.04 | 74.93±0.15 | 67.30±0.02 | 69.29±0.10 | 61.12±0.04 |
| EASE | 57.99±1.15 | 45.36±1.34 | 58.18±0.98 | 46.21±1.33 | 59.86±0.78 | 47.53±0.92 | 81.75±0.32 | 76.20±0.19 | 81.18±0.20 | 74.62±0.09 | 76.51±0.31 | 68.67±0.24 |
| COFiMA | 57.70±0.86 | 47.60±0.78 | 58.43±0.75 | 48.12±1.04 | – | – | 82.05±0.15 | 76.43±0.12 | 81.54±0.16 | 75.15±0.14 | – | – |
| MOS | 67.71±0.62 | 57.14±0.92 | 65.10±0.44 | 54.25±0.32 | 63.72±0.65 | 51.22±0.45 | 82.59±0.34 | 77.80±0.20 | 81.32±0.22 | 75.62±0.12 | 75.22±0.15 | 67.18±0.08 |
| **MIN (Ours)** | **72.89±0.45** | **64.32±0.89** | **72.37±0.60** | **63.66±0.26** | **66.15±0.35** | **57.08±0.27** | **85.18±0.19** | **79.75±0.07** | **83.69±0.17** | **78.08±0.15** | **82.26±0.16** | **75.72±0.19** |

## 5.1 Experimental settings

**Datasets.** We conduct experiments on six benchmark datasets, including CIFAR100 [47], CUB200 [48], ImageNet-A [49], ImageNet-R [50], FOOD101 [51] and Omnibenchmark [52]. There are 100 categories in CIFAR100, 101 categories in FOOD101, 200 categories in CUB200, ImageNet-A, ImageNet-R, and 300 categories in Omnibenchmark. Detailed information for the datasets is shown in the supplementary.

**Protocols.** Following recent CIL works, we split a dataset into $T$ tasks, denoted as $T$ steps. Each task learns the same number of categories. To further investigate the performance of all methods across various step sizes, we set the range of $T$ from 5 to 50, i.e., $T = 5$, 10, 20, and 50. Following [37, 34, 39], the learning order is determined by the random seed 1993. We run each experiment 3 times and report the average result. Additionally, we also report the results using other random seeds in the supplementary.

**Evaluation metric.** After learning the task $t$, we test the model on all known categories. Assuming that $A_t$ represents the accuracy of the model after $t$ tasks. We use two metrics, the average accuracy across $T$ tasks $\bar{A} = \frac{1}{T}\sum_{t=1}^{T} A_t$ and the last task accuracy $A_T$, to measure the CIL performance.

**Comparison methods.** We choose state-of-the-art PTM-based CIL methods for comparison. To be specific, we choose L2P [30], DualPrompt [31], CODA-Prompt [32], SLCA [33], FeCAM [35], RanPAC [36], APER [34], EASE [37], COFiMA [38] and MOS [39]. In addition, we also choose ACIL [26] as the baseline of our work.

**Implementation details.** Since the wide range of PTMs are publicly accessible, we choose two representative models following [31, 37, 34, 39], denoted as **ViT-B/16-IN21K** [53] and **ViT-B/16-IN1K**. They are both initially pre-trained on ImageNet-21K, while the latter is further finetuned on ImageNet-1K. In MIN, we set the batch size to 128 and train for 10 epochs using SGD optimizer with momentum. The learning rate is initially set to 0.001 and decays to 0 following a cosine annealing

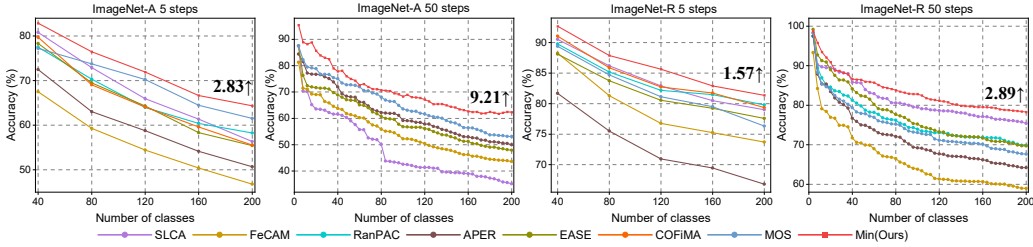

Figure 4: The performance of each learning session under different settings of ImageNet-A and ImageNet-R. All methods are initialized with **ViT-B/16-IN1K**.

Table 3: Average and last performance comparison on FOOD-101 and Omnibenchmark datasets with **ViT-B/16-IN21K** as the backbone. We report all compared methods with their source code. The best performance is shown in bold, and the second-best result is underlined. All methods are implemented without using exemplars.

| Methods | FOOD-101 | | | | | | Omnibenchmark | | | | | |
| | 10 steps | | 20 steps | | 50 steps | | 10 steps | | 20 steps | | 50 steps | |
| | $\overline{A}$ | $A_T$ | $\overline{A}$ | $A_T$ | $\overline{A}$ | $A_T$ | $\overline{A}$ | $A_T$ | $\overline{A}$ | $A_T$ | $\overline{A}$ | $A_T$ |
|---|---|---|---|---|---|---|---|---|---|---|---|---|
| L2P | $82.92_{\pm0.65}$ | $74.93_{\pm0.85}$ | $78.74_{\pm0.54}$ | $70.66_{\pm0.52}$ | $59.60_{\pm0.38}$ | $50.26_{\pm0.26}$ | $73.01_{\pm0.33}$ | $63.29_{\pm0.26}$ | $71.65_{\pm0.34}$ | $61.60_{\pm0.16}$ | $69.54_{\pm0.47}$ | $59.05_{\pm0.22}$ |
| DualPrompt | $86.25_{\pm0.45}$ | $80.02_{\pm0.40}$ | $80.48_{\pm0.37}$ | $72.78_{\pm0.26}$ | $67.23_{\pm0.58}$ | $58.80_{\pm0.67}$ | $77.68_{\pm0.18}$ | $67.45_{\pm0.45}$ | $74.13_{\pm0.35}$ | $64.88_{\pm0.27}$ | $71.28_{\pm0.56}$ | $59.93_{\pm0.38}$ |
| CODA-Prompt | $88.91_{\pm0.33}$ | $83.14_{\pm0.15}$ | $83.19_{\pm0.30}$ | $76.53_{\pm0.26}$ | $62.84_{\pm1.77}$ | $51.57_{\pm0.57}$ | $79.18_{\pm0.42}$ | $70.01_{\pm0.65}$ | $74.85_{\pm0.18}$ | $65.78_{\pm0.22}$ | $70.50_{\pm0.14}$ | $60.50_{\pm0.30}$ |
| ACIL | $90.68_{\pm0.03}$ | $86.03_{\pm0.07}$ | $90.66_{\pm0.08}$ | $85.81_{\pm0.07}$ | $90.34_{\pm0.04}$ | $85.38_{\pm0.05}$ | $84.26_{\pm0.35}$ | $76.11_{\pm0.52}$ | $84.76_{\pm0.26}$ | $75.96_{\pm0.47}$ | $84.28_{\pm0.18}$ | $75.31_{\pm0.44}$ |
| SLCA | $91.10_{\pm0.37}$ | $86.40_{\pm0.17}$ | $89.61_{\pm0.27}$ | $83.42_{\pm0.08}$ | $86.08_{\pm0.16}$ | $78.69_{\pm0.12}$ | $82.46_{\pm0.26}$ | $74.24_{\pm0.19}$ | $80.55_{\pm0.44}$ | $70.77_{\pm0.75}$ | $78.48_{\pm0.21}$ | $69.14_{\pm0.15}$ |
| FeCAM | $90.87_{\pm0.50}$ | $86.56_{\pm0.42}$ | $90.75_{\pm0.32}$ | $86.05_{\pm0.31}$ | $90.08_{\pm0.24}$ | $85.21_{\pm0.11}$ | $83.74_{\pm0.16}$ | $77.18_{\pm0.22}$ | $83.02_{\pm0.18}$ | $76.19_{\pm0.16}$ | $83.01_{\pm0.26}$ | $75.58_{\pm0.31}$ |
| RanPAC | $92.13_{\pm0.32}$ | $88.90_{\pm0.15}$ | $91.61_{\pm0.20}$ | $87.16_{\pm0.15}$ | $91.07_{\pm0.16}$ | $86.64_{\pm0.09}$ | $85.04_{\pm0.12}$ | $78.83_{\pm0.18}$ | $84.14_{\pm0.08}$ | $76.77_{\pm0.05}$ | $84.16_{\pm0.10}$ | $76.33_{\pm0.05}$ |
| APER | $88.49_{\pm0.19}$ | $83.60_{\pm0.05}$ | $88.49_{\pm0.18}$ | $83.19_{\pm0.05}$ | $89.03_{\pm0.08}$ | $83.38_{\pm0.04}$ | $80.72_{\pm0.20}$ | $74.34_{\pm0.14}$ | $80.26_{\pm0.21}$ | $73.36_{\pm0.16}$ | $80.27_{\pm0.08}$ | $73.00_{\pm0.10}$ |
| EASE | $88.63_{\pm0.25}$ | $83.26_{\pm0.22}$ | $85.27_{\pm0.28}$ | $79.30_{\pm0.35}$ | $85.61_{\pm0.30}$ | $78.75_{\pm0.33}$ | $75.71_{\pm0.45}$ | $68.59_{\pm0.40}$ | $74.99_{\pm0.37}$ | $65.72_{\pm0.25}$ | $73.56_{\pm0.48}$ | $65.52_{\pm0.37}$ |
| COFiMA | $90.83_{\pm0.06}$ | $85.84_{\pm0.08}$ | $88.12_{\pm0.15}$ | $81.82_{\pm0.16}$ | – | – | $81.69_{\pm0.40}$ | $73.38_{\pm0.28}$ | $80.30_{\pm0.42}$ | $71.18_{\pm0.25}$ | – | – |
| MOS | $92.54_{\pm0.11}$ | $89.27_{\pm0.08}$ | $92.02_{\pm0.18}$ | $88.29_{\pm0.22}$ | $91.57_{\pm0.10}$ | $87.08_{\pm0.08}$ | $86.10_{\pm0.11}$ | $80.12_{\pm0.15}$ | $85.22_{\pm0.08}$ | $78.58_{\pm0.12}$ | $85.06_{\pm0.12}$ | $77.49_{\pm0.15}$ |
| **MIN (Ours)** | $\mathbf{93.36}_{\pm0.07}$ | $\mathbf{90.04}_{\pm0.06}$ | $\mathbf{93.20}_{\pm0.12}$ | $\mathbf{89.55}_{\pm0.16}$ | $\mathbf{93.60}_{\pm0.07}$ | $\mathbf{89.47}_{\pm0.12}$ | $\mathbf{87.36}_{\pm0.04}$ | $\mathbf{80.55}_{\pm0.10}$ | $\mathbf{87.79}_{\pm0.03}$ | $\mathbf{79.90}_{\pm0.11}$ | $\mathbf{87.58}_{\pm0.15}$ | $\mathbf{79.35}_{\pm0.19}$ |

decay pattern. The dimension $d_2$ of the latent vector within the Pi-Noise layer is set to 192, with more configurations of this hyperparameter detailed in the supplementary materials.

## 5.2 Comparison with State-of-the-arts

In this section, we compare the proposed MIN with other state-of-the-art methods on six benchmark datasets and different pre-trained weights. Complete comparison experiments are provided in the supplementary.

**CIFAR & CUB.** Tab. 1 presents the performance comparison of different methods using ViT-B/16-IN21K on the CIFAR100 and CUB200 datasets. We report the results under three different settings for each dataset: 10 steps, 20 steps, and 50 steps. As the number of tasks increases, the incremental learning difficulty progressively intensifies. Due to the lack of inter-task information guidance, all methods exhibit a significant performance decline in the 50-steps compared to the 10-steps scenario. The proposed method achieves the best performance across all three settings, with its advantage over the runner-up method becoming notably amplified under the 50-steps setting. Additionally, Fig. 3 presents the incremental performance trends of different methods using ViT-B/16-IN1K. For each dataset, we report results under both 5-steps and 50-steps settings to demonstrate the impact of the number of tasks on incremental performance. As shown in Fig. 3, the proposed method exhibits a more gradual decline rate in the 50-steps setting compared to other approaches. When contrasting the 5-step and 50-step results on CIFAR100, the performance advantage of our method over the runner-up approach increases from 0.76% in the 10-steps setting to 4.28% under 50-steps settings. Similarly, for CUB200, the performance gap expands from 0.86% to 2.12% between these two settings.

**ImageNet-A/R.** Tab. 2 presents the performance comparison of different methods using ViT-B/16-IN21K on the ImageNet-A/R datasets, with results reported under three different settings. The proposed method demonstrates superior performance across all settings, achieving a 6%–9% improvement in final accuracy over the runner-up approach on ImageNet-A, and a 2%–3% enhancement on ImageNet-R. Furthermore, Fig. 4 illustrates the incremental trends of various methods with

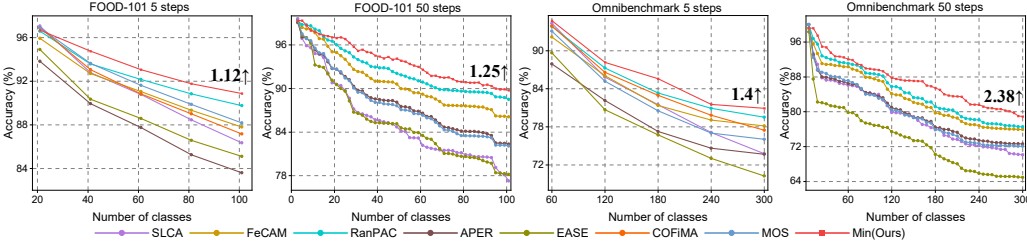

Figure 5: The performance of each learning session under different settings of FOOD-101 and Omnibenchmark. All methods are initialized with **ViT-B/16-IN1K**.

Table 4: Results of ablation study. NE+$\bar{\varepsilon}$ calculates the average noise of all tasks. NE+$\varepsilon_\mu$ only uses the $\mu$ of each task noise. NE+$\varepsilon_\sigma$ only uses the $\sigma$ of each task noise. NE+$\varepsilon_t$ uses the last task noise and NE+$\varepsilon_i$ uses a random task noise.

| Methods | CIFAR | | CUB | | IN-A | | IN-R | | FOOD | | Omni. | |
|---|---|---|---|---|---|---|---|---|---|---|---|---|
| | $\overline{A}$ | $A_T$ | $\overline{A}$ | $A_T$ | $\overline{A}$ | $A_T$ | $\overline{A}$ | $A_T$ | $\overline{A}$ | $A_T$ | $\overline{A}$ | $A_T$ |
| Baseline | 91.18±0.03 | 86.82±0.07 | 91.55±0.16 | 87.08±0.17 | 62.23±0.45 | 53.65±0.57 | 75.14±0.05 | 68.45±0.27 | 90.26±0.03 | 85.57±0.10 | 84.55±0.33 | 76.82±0.44 |
| w/ NE+$\bar{\varepsilon}$ | 94.07±0.04 | 90.22±0.10 | 92.89±0.09 | 90.77±0.15 | 71.43±0.55 | 62.59±0.70 | 83.86±0.24 | 78.65±0.11 | 92.67±0.05 | 89.34±0.09 | 86.77±0.15 | 80.12±0.18 |
| w/ NE+$\varepsilon_\mu$ | 89.27±0.24 | 85.63±0.33 | 88.97±0.46 | 85.45±0.45 | 65.87±1.27 | 55.36±1.02 | 76.27±0.44 | 67.15±0.34 | 88.45±0.09 | 83.16±0.12 | 82.15±0.68 | 73.95±0.70 |
| w/ NE+$\varepsilon_\sigma$ | 94.37±0.05 | 91.32±0.12 | 93.15±0.17 | 90.47±0.09 | 70.45±0.25 | 61.32±0.33 | 84.15±0.20 | 79.05±0.12 | 92.90±0.05 | 89.25±0.04 | 87.06±0.07 | 80.12±0.05 |
| w/ NE+$\varepsilon_t$ | 92.36±0.18 | 87.92±0.27 | 92.37±0.08 | 87.25±0.12 | 63.23±0.75 | 54.15±0.92 | 79.15±0.16 | 71.71±0.28 | 90.62±0.12 | 86.12±0.05 | 85.02±0.06 | 77.47±0.08 |
| w/ NE+$\varepsilon_i$ | 92.65±0.21 | 88.10±0.15 | 92.15±0.23 | 88.76±0.26 | 64.39±1.08 | 54.88±1.35 | 81.39±0.65 | 73.15±0.81 | 91.08±0.30 | 86.57±0.19 | 84.77±0.24 | 77.96±0.16 |
| MIN | 95.12±0.05 | 92.12±0.16 | 94.00±0.15 | 91.22±0.18 | 72.89±0.45 | 64.32±0.89 | 85.18±0.19 | 79.75±0.07 | 93.36±0.07 | 90.04±0.06 | 87.36±0.04 | 80.55±0.10 |

ViT-B/16-IN1K, where our approach shows amplified advantages against the suboptimal method as task number increases.

**FOOD101 & Omnibenchmark** Tab. 3 demonstrates the performance comparison of various methods using ViT-B/16-IN21K on the FOOD101 and Omnibenchmark datasets. The proposed method maintains superior performance in all three settings. Fig. 5 illustrates the incremental trends with ViT-B/16-IN1K, where the findings further validate the progressive widening of the performance gaps between our method and the runner-up approach as the number of tasks increases.

### 5.3 Ablation Study and Visualizations

**Ablation study.** To verify the effectiveness of each component in MIN, we use the ViT-B/16-IN21K to conduct an ablation study on all six datasets with the 10-steps setting. **Baseline** denotes the approach using analytic classifier with the pre-trained backbone [26](see Sec. 3). It should be noted that in the absence of *Noise Mixture*, the mixing module operates by computing the average of all noises. Additionally, instead of employing an auxiliary classifier for training, it directly utilizes $W_t$ to calculate the cross-entropy loss to update $P_t$. The results are presented in Tab. 4 which validate the effectiveness of each strategy.

**Visualizations.** We use Grad-CAM visualization to further demonstrate how MIN applies beneficial noise to improve the performance of baseline methods in Fig. 2. The comparative results demonstrate that while the baseline method also focuses on the primary subject, its activation region exhibits greater dispersion with pronounced responses in irrelevant regions. For example, in the first set of images labeled vase, the baseline method generates strong activation for the background wall and the hanging painting. But the proposed method pays more attention to the main part of the vase, and suppresses the irrelevant background information. This indicates that the proposed method works on improving the accuracy by suppressing invalid features. We provide more visualizations in the supplementary.

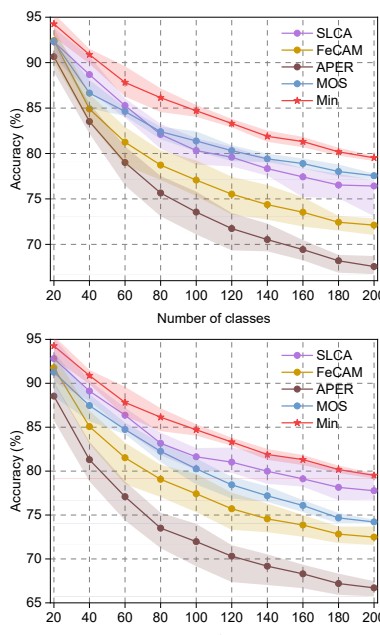

Figure 6: Incremental trends with random orders on the ImageNet-R dataset using ViT-B/16-IN21K (Top) and ViT-B/16-IN1K (Bottom).

**Random seed experiments.** We shuffle the learning order with the random seeds, i.e., {1993, 1994, 1995, 1996, 1997, 1998}. Therefore, we obtain six results of different learning order for each method on ImageNet-R using ViT-B/16-IN21K and ViT-B/16-IN1K. The results are shown in Fig. 6. For each method, the solid line indicates the mean of the five outcomes and the shading indicates the fluctuation range. The results demonstrate that MIN consistently outperforms other methods by a substantial margin.

**Robustness of hyperparameters.** There are two hyperparameters in the baseline method, i.e., the coefficient $\gamma$ of regularization terms and the buffer size. We investigate the robustness by changing these hyperparameters. Specifically, we select $\gamma$ from {10, 50, 100, 500, 1000} and the buffer size

from {1024, 2048, 4096, 8192, 16384}. We report the average accuracy in Fig. 7(a). The results indicate that selecting $\gamma$ as 100 or 500, along with a buffer size of 8196 or 16384, yields better performance. In addition, we change the dimension of the hidden layer $P_t$ to investigate its impact on the model performance. Specifically, we select from {96, 192, 256, 384, 512, 768} as the dimension that $W_{\text{down}}$ reduce the input feature into. And we report the incremental trends in Fig. 7(b). Then, the incremental trends of the number of trainable parameters are shown in Fig. 7(c). As shown in the figure, the model achieves the best performance when the dimension is set to 192 (dim=192), and further increasing this dimension results in performance degradation. Hence, we adopt dim=192 as the dimension of the hidden layer in $P_t$. In addition, Tab. 5 shows the results of changing $\tau$ in Eq. (12). It indicates that the proposed method is not sensitive for $\tau$.

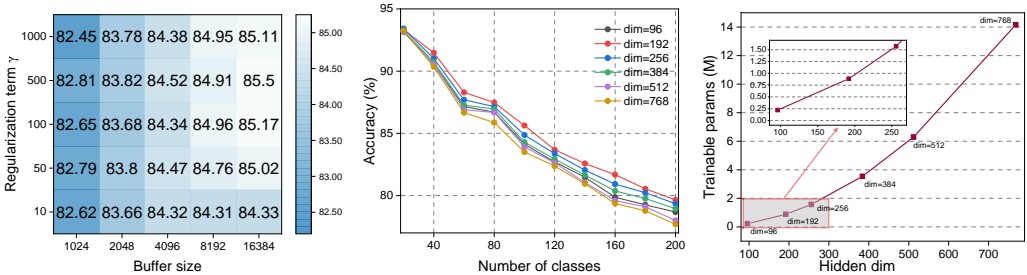

Figure 7: Further analysis on parameter robustness.

**Additional experiments.** We set the hyperparameter $d_2$ to 192, which is the same as in the comparative experiments, to analyze the number of model parameters. From Fig. 8(a) and (b), it can be seen that the total number of model parameters of MIN is the same as most of the methods, while the number of learnable parameters for new tasks is less than most of the methods. Fig. 8(c) shows the impact of Pi-Noise compared to alternatives including Adapter [54], VPT-shallow [55] and

Table 5: Robustness of $\tau$ in the setting of ImageNet-R 10 steps.

| $\tau$ | $\overline{A}$ | $A_T$ |
|---|---|---|
| 0.50 | $84.92_{\pm 0.21}$ | $79.30_{\pm 0.23}$ |
| 1.00 | $84.79_{\pm 0.22}$ | $79.38_{\pm 0.10}$ |
| 1.50 | $84.91_{\pm 0.19}$ | $79.59_{\pm 0.12}$ |
| 2.00 | $85.18_{\pm 0.19}$ | $79.75_{\pm 0.07}$ |

VPT-deep [55]. Following ACIL [26], we use these alternative approaches to train the model in the first task and then learn subsequent tasks by using analytic classifier. The comparison demonstrates that Pi-Noise shows superiority over other methods.

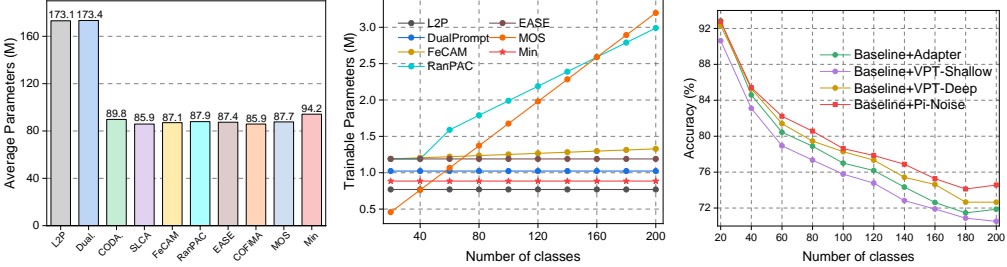

Figure 8: More experiments results. (a) presents the average number of model parameters for different methods across 10 steps. (b) presents the number of trainable parameters for different methods across 10 steps. (c) Impact of Pi-Noise compared to alternatives under the setting of ImagenNet-R 10 steps.

# 6 Conclusion

Incremental learning is a key step to achieve a continual AI system. This paper propose mixture of positive-incentive noise (MIN) for class incremental learning based on pre-trained models. Specifically, the proposed method expands a Pi-Noise generator for each task to adapt new tasks without the degradation of backbone generalization. Considering the cross-task collaboration among different Pi-Noises, we guide the model via an auxiliary classifier to learn a set of adaptive weights for adjusting their mixing ratios. Extensive experiments validate the effectiveness of MIN.

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

# Appendix

## A    Overview

In this supplementary material, we provide more details about MIN. Firstly, we give the implementation details in Sec. B. Then, we provide the details of six datasets in Sec. C. In Sec. D, we provide hardware information used for all experiments. Then, we provide descriptions of all comparison methods in Sec. E. In Sec. F, we provide more visualizations to further illustrate the effectiveness of MIN. In the end, we provide more comparison experimental results in Sec. G. Limitations of MIN is provided in Sec. H.

## B    Implementation details

We run all the experiments with PyTorch [56] and reproduce all other comparison methods with Pilot [57]. Following [30, 32, 36, 34, 39], we conduct experiments with the ViT-B/16-IN21K and ViT-B/16-IN1K. In MIN, we train the model using the SGD optimizer, with a batch size of 128 for 10 epochs. The learning rate decays from 0.001 with cosine annealing to zero. There are subtle differences in these hyper-parameter between different datasets due to scale of datatsets. We set the dimension of hidden layer as 192 according to Sec. 5.3. The hyperparameters of baseline method, i.e., regularization term $\gamma$ and buffer size, are set to 100 and 16384 according to Sec. 5.3.

## C    Datasets

The details of six datasets are illustrated in Tab. 6, including CIFAR100, CUB, ImageNet-A/R, FOOD101 and Omnibenchmark.

Table 6: Details of six datasets.

| Datasets | Classes | Train | Test | Avg size |
|---|---|---|---|---|
| CIFAR-100 | 100 | 50,000 | 10,000 | 32×32 |
| CUB-200 | 200 | 9,430 | 2,358 | 467×386 |
| ImageNet-A | 200 | 5,960 | 1,515 | 443×427 |
| ImageNet-R | 200 | 24,000 | 6,000 | 443×427 |
| Omnibenchmark | 300 | 89,697 | 5,985 | 764×581 |
| FOOD-101 | 101 | 75,750 | 25,250 | 496×475 |

## D    Hardware Information

The hardware information is as follows:

- **CPU**: Intel Xeon(R) Gold 6244 CPU
- **GPU**: 2×NVIDIA GeForce RTX 4090
- **Mem**: 8×DDR4 SAMSUNG-32GB

## E    Comparison methods

We totally choose the **10** approaches for comparison. A brief description of these methods is given below.

**L2P.**    L2P [30] freezes the parameters of pre-trained weights and uses visual prompt tuning to learn the new tasks. It constructs a prompt pool and selects a suitable prompt for each sample with a key-value mapping strategy.

**DualPrompt.**    DualPrompt [31] is the improvement work for L2P. It divides the prompt into two types, i.e., general prompts and expert prompts.

**CODA-Prompt.** CODA-Prompt [32] replaces prompt reweighting with an attention-based prompt recombination during the prompt selection process. Owning to the attention module, it needs more parameters and training time.

**SLCA.** SLCA [33] updates the backbone slowly and rectifies the classifier with pseudo features which are sampled from the Gaussian distribution modeled by the prototype of old categories.

**FeCAM.** FeCAM [35] applies the Mahalanobis distance to replace the Euclidean distance for classification without updating the backbone. In addition, it shows that modeling the feature covariance relations is better than previous attempts at sampling features from normal distributions and training a linear classifier.

**RanPAC.** RanPAC [36] introduces a frozen random projection layer between the feature output of the pre-trained model and the classifier, combines with a nonlinear activation function, expands the feature dimension and enhances linear separability, so as to alleviate the forgetting problem without updating the backbone.

**APER.** APER [34] builds the prototype-based classifier and uses a cosine classifier for classification. It adapts the new data only in the first task with additional modules. We choose the best version, i.e., APER-adapter for comparison.

**EASE.** EASE [37] is based on the idea of network expansion. It adds a new branch to each transformer block for each task. Although it trains only one branch in training process, it needs to infer multiple times through the backbone for test.

**COFiMA.** COFiMA [38] integrates the model parameters of the current task and the previous task by introducing the Fisher information weighting mechanism and balances the plasticity and stability of the model.

**MOS.** MOS [39] alleviates the parameter drift problem through adapter merging technology, and combines the self-optimization retrieval mechanism without training to dynamically correct the module matching error, which solves the catastrophic forgetting problem in the incremental learning of pre-trained models from two perspectives of parameters and retrieval.

## F    More Visualizations

We provide more visualizations in Fig. 9 to demonstrate the effectiveness of MIN. For instance, three bird species from the CUB dataset are shown, which require fine-grained recognition for accurate differentiation. The backbone network of the baseline method fails to learn cross-task fine-grained features, only achieving coarse-grained discrimination with limited activation values concentrated on the birds' main body regions. In contrast, MIN learns task-specific Pi-Noise that suppresses common features shared across bird species while generating stronger activation responses to discriminative fine-grained characteristics. This indicates that MIN can effectively inhibit the backbone network's activation of common features across similar categories in different tasks, thereby facilitating the learning of underlying target features and avoiding confusion of decision boundaries between multiple tasks.

## G    More Experimental Results with SOTAs

In this section, we show more experimental results of different methods. As demonstrated from Fig. 10 to Fig. 21, we report the incremental performance of different methods with ViT-B/16-IN21K and ViT-B/16-IN1K on six datasets. MIN consistently outperforms other methods on different datasets by a substantial margin.

## H    Limitations

MIN relies on an excellent feature extractor to provide a prior information to guide Pi-Noise generator for adaptation of subsequent downstream tasks. With the rapid development of pre-training technology, the weights of the backbone network trained by self-supervised learning on large-scale unlabeled data are easy to access, which reduces this limitations. In addition, MIN requires an additional downsampling layer in each Pi-Noise layer (although they are training-free). Introduces additional

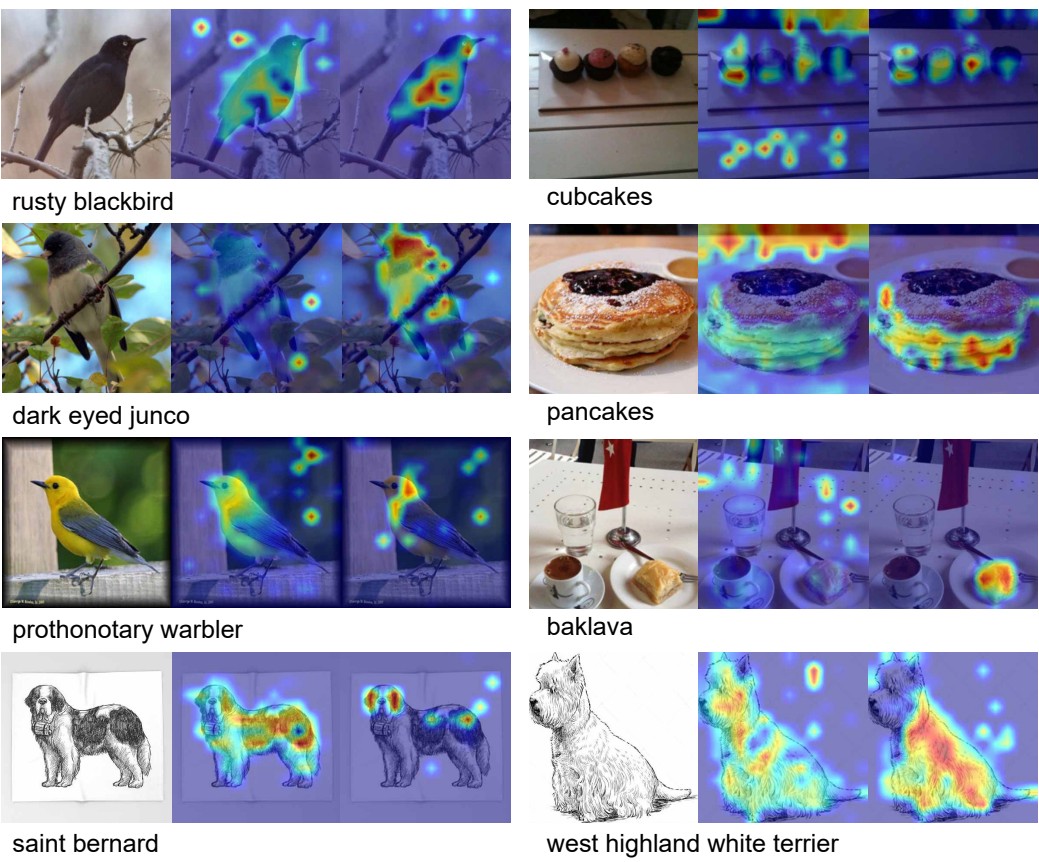

rusty blackbird

cubcakes

dark eyed junco

pancakes

prothonotary warbler

baklava

saint bernard

west highland white terrier

Figure 9: More visualizations

network parameters (equivalent to 4% of the number of parameters for ViT-B/16). However, the number of actual training parameters is much smaller (equivalent to 0.8M parameters). Therefore, it is worth trade-off for the simplicity of implementation and low-cost training.

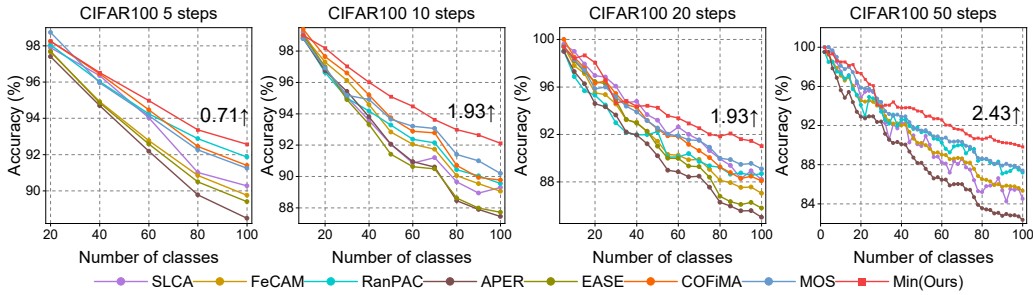

Figure 10: Incremnetal trends of different approaches using **ViT-B/16-IN21K** on the CIFAR100 dataset.

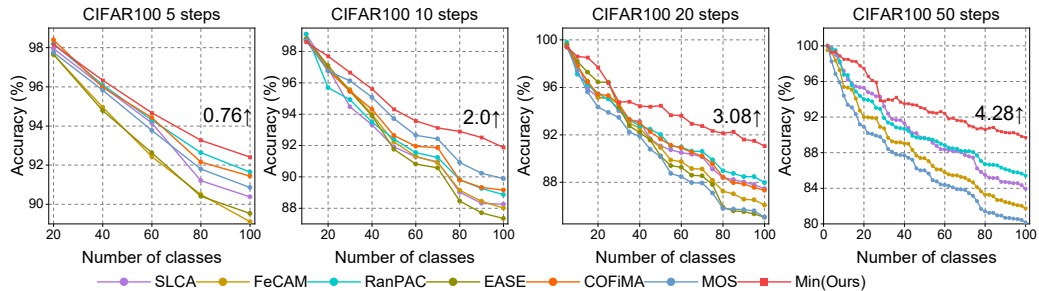

Figure 11: Incremntal trends of different approaches using **ViT-B/16-IN1K** on the CIFAR100 dataset.

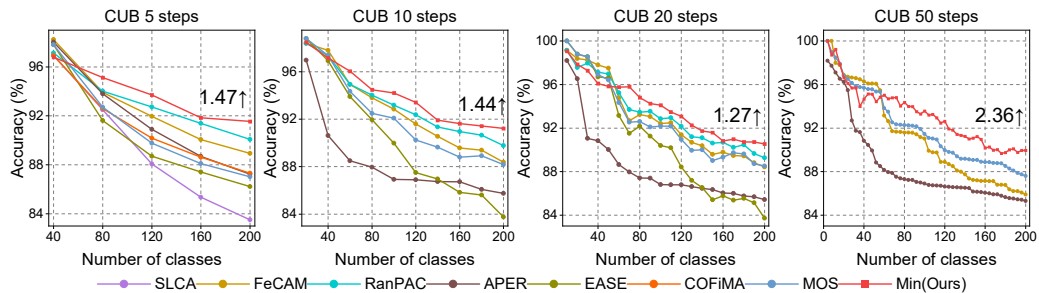

Figure 12: Incremntal trends of different approaches using **ViT-B/16-IN21K** on the CUB dataset.

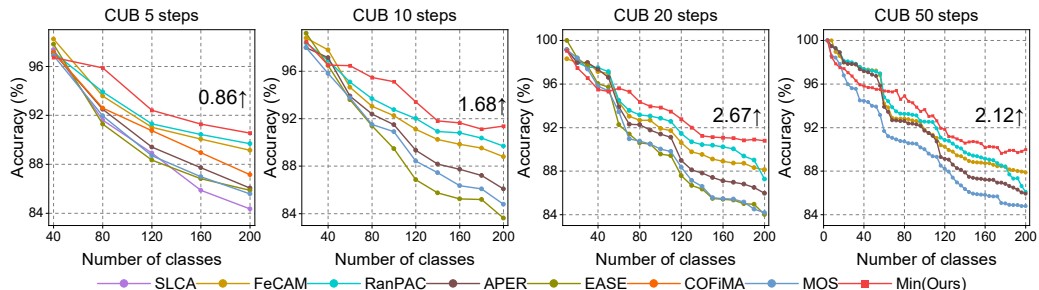

Figure 13: Incremntal trends of different approaches using **ViT-B/16-IN1K** on the CUB dataset.

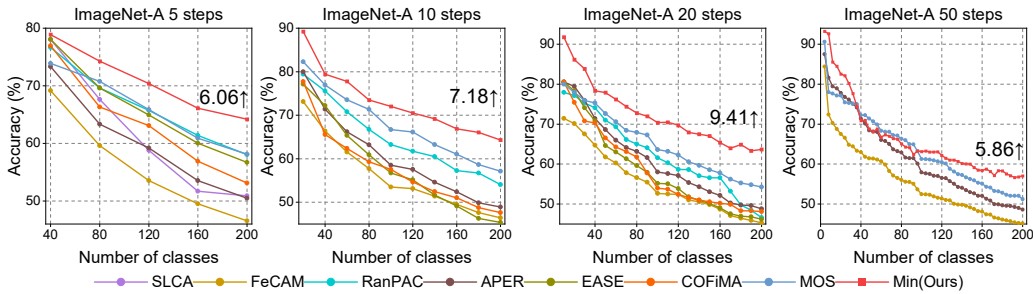

Figure 14: Incremntal trends of different approaches using **ViT-B/16-IN21K** on the ImageNet-A dataset.

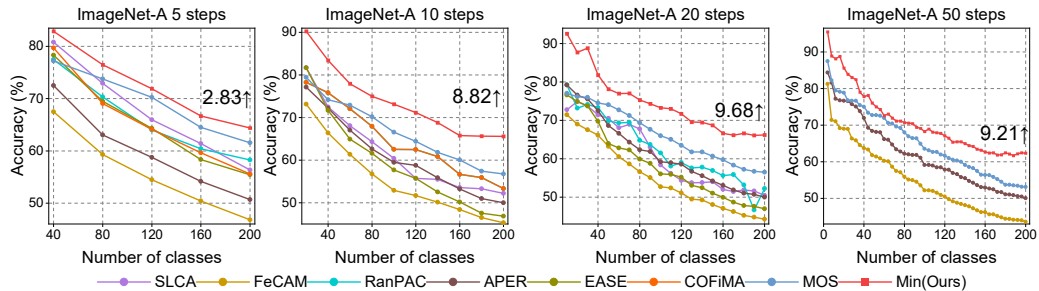

Figure 15: Incremnetal trends of different approaches using **ViT-B/16-IN1K** on the ImageNet-A dataset.

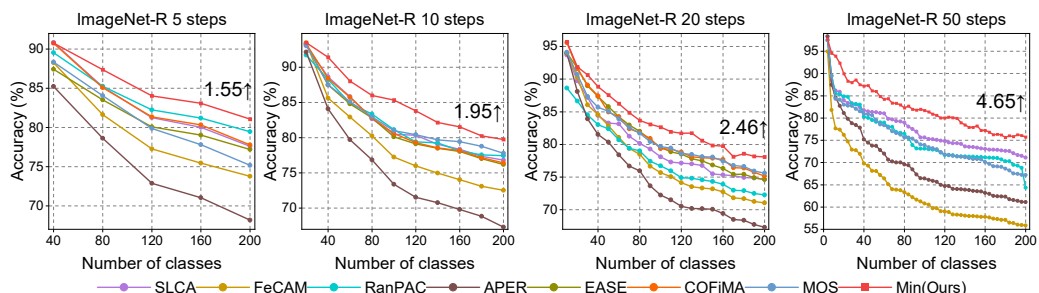

Figure 16: Incremnetal trends of different approaches using **ViT-B/16-IN21K** on the ImageNet-R dataset.

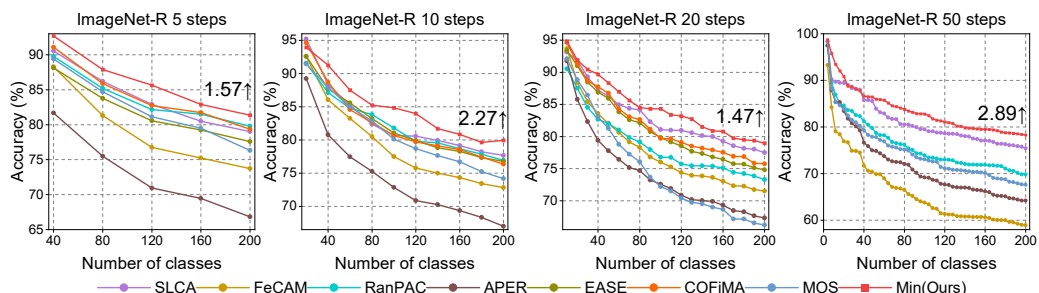

Figure 17: Incremnetal trends of different approaches using **ViT-B/16-IN1K** on the ImageNet-R dataset.

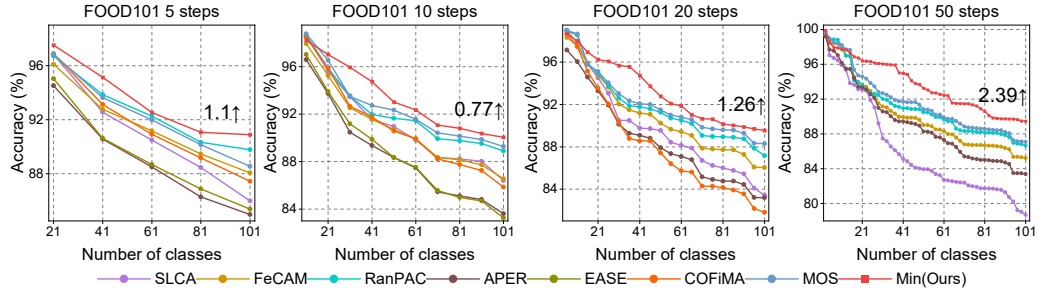

Figure 18: Incremnetal trends of different approaches using **ViT-B/16-IN21K** on the FOOD101 dataset.

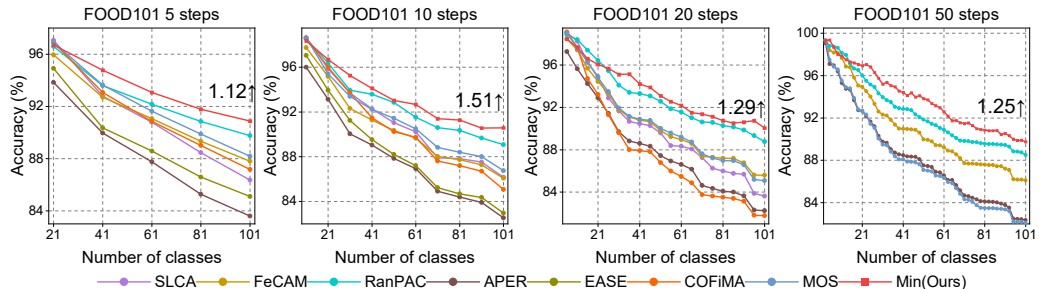

Figure 19: Incremnetal trends of different approaches using **ViT-B/16-IN1K** on the FOOD101 dataset.

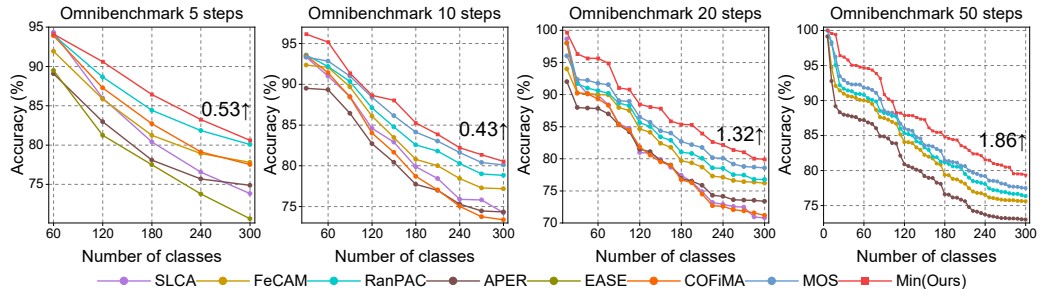

Figure 20: Incremnetal trends of different approaches using **ViT-B/16-IN21K** on the Omnibenchmark dataset.

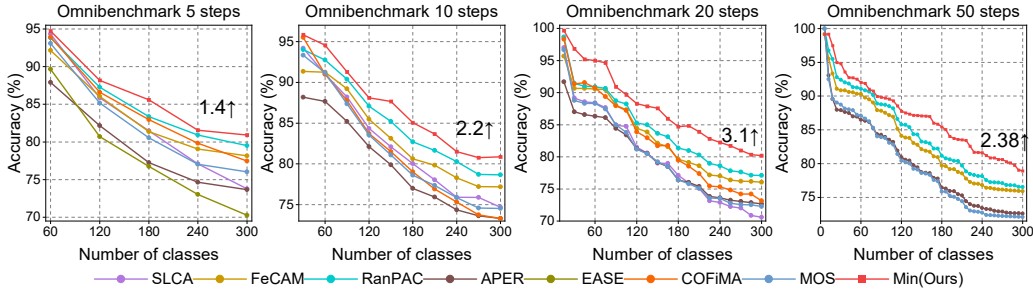

Figure 21: Incremnetal trends of different approaches using **ViT-B/16-IN1K** on the Omnibenchmark dataset.

