# OpenReview forum: "Mixture of Noise for Pre-Trained Model-Based Class-Incremental Learning"
_NeurIPS.cc/2025/Conference — NeurIPS 2025 poster_

### Official Review · Reviewer_wYxB · 2025-06-21

**Clarity:** 2
**Significance:** 2
**Originality:** 3
**Rating:** 4
**Confidence:** 4

**Summary:**

The paper presents Mixture of Noise (MIN), a new approach to exemplar-free class-incremental learning that reframes catastrophic forgetting as "parameter drift" (essentially noise that corrupts previous task knowledge). Guided by this perspective, MIN explicitly learns beneficial noise for each new task and injects it into intermediate features to mask spurious patterns. An auxiliary classifier blends these different noise patterns using adaptive weights, allowing the system to handle multiple tasks seamlessly. Tested across six benchmarks including CIFAR-100, CUB-200, ImageNet variants, and Food-101, MIN achieves state-of-the-art performance.

**Questions:**

The authors are asked to answer all the questions in Weaknesses, including (but not limited only to):

1. Can the authors provide the formal definition of "non-essential features", followed by theoretical justification (or at least empirical evidence) to clarify how the non-essential features are assimilated into decision boundaries?

2. Can the authors provide the formal definition of "task-specific redundancy" in the context of pre-trained models, followed by theoretical justification (or at least empirical evidence) to clarify how they distinguish between features that are irrelevant for future (new) tasks versus features that are irrelevant for the current task?

3. The noise-mixing weights depend on the temperature $\tau$ (Eq. 12) and the choice of prototype dimension. How sensitive are results to $\tau$ or to the dimensionality $d_2$ used in the noise generator? Have you tuned these, and is performance stable across reasonable values? Providing an ablation (or variance over seeds) would clarify robustness.

4. Did you ensure all baselines use identical backbones, optimization hyperparameters, and exemplar-free settings? For example, in original paper MOS for CUB dataset and 10 tasks the average accuracy reported 93.40 but, in your work, it is 92.08.

5. You use a weighted sum of past noises (Eq. 13). Did you compare this to simpler strategies, such as averaging or using only the current task’s noise? The ablation shows both steps, but it might help to see intermediate variants (e.g. mixing without learning weights vs. with random weights) to fully justify the learned mixture

**Ethical Concerns:**

["NO or VERY MINOR ethics concerns only"]

**Final Justification:**

I appreciate the authors' rebuttal to my initial comments and responses to my follow up questions. I am still not convinced that the novelty of this paper is significant enough. Nevertheless, as majority of my concerns have been addressed, I have revised my score.

**Limitations:**

The authors do not explicitly discuss limitations. But, MIN has so far only been validated on image classification benchmarks with ViT backbones; its effectiveness in other domains or other tasks is unknown. Moreover, any assumptions behind “beneficial noise” (e.g. tasks share common distractors) should be clarified.

**Paper Formatting Concerns:**

No significant deviations from NeurIPS formatting were found.

**Quality:**

3

**Strengths And Weaknesses:**

**Strengths:**

1. Recasting forgetting as noise and injecting learned “positive-incentive” noise is a new approach. The paper outlines two core strategies: Noise Expansion (one noise generator per task) and Noise Mixture (learning weights to combine them). Figures and equations provide a coherent view of the training pipeline (e.g., Fig.1 shows the Pi-Noise layer and three-step training).
2. The experiments are reasonably comprehensive. MIN is tested on six datasets and multiple incremental settings (10/20/50 steps). Across these, MIN outperforms baselines like L2P, DualPrompt, CODA-Prompt, ACIL, FeCAM, RanPAC, etc.
3. The authors emphasize that only a small, fixed number of parameters are learned per task. Figures 6(a)-(b) (described in Sec.5.3) show that MIN’s total parameters are comparable to other methods, while new-task parameters are lower than most.

**Weaknesses:**

1. The authors make many unclear statements in the introduction regarding the challenges of using pre-trained models (PTMs) in incremental learning, without providing support or evidence. Some notable ones:

1.1 “Pre-trained models inherently exhibit task-specific redundancy”: It is unclear what is meant by "task-specific redundancy". Does this mean that pre-trained models carry features irrelevant to future tasks or that some learned features are irrelevant even for the current task?

1.2 “Models inadvertently assimilate non-essential features into the decision boundaries”: The term "non-essential features" is ambiguous and needs further clarification. Also, the mechanism by which non-essential features are "assimilated" into decision boundaries is vague and lacks supporting evidence.

1.3 “These features induce catastrophic forgetting through two mechanisms”: The two mechanisms are not clearly distinguished. While catastrophic forgetting often refers to the forgetting the previously learned tasks, it appears that the authors conflate catastrophic forgetting with inter-task confusion, treating both as the same: The performance reduction due to task confusion in future tasks should not be categorized as forgetting. Additionally, forgetting is generally attributed to parameter interference or representational overlap, rather than merely feature redundancy.

1.4 “This issue stems from the parameter sensitivity of PTMs”: The term "parameter sensitivity" is not defined. Does it refer to high sensitivity to perturbations, fine-tuning instability, or something else?

1.5 “It is advisable to mitigate individual tasks’ dependence on redundant features”: It’s not clear how one would determine or mitigate “dependence on redundant features”. This claim is too general to be feasible.

1.6 “Catastrophic forgetting stems from parameter drift for old tasks, it can be conceptualized as the intrusion of noise during learning a new task, which undermining the original pattern of the old framework.” The reviewer is not convinced about this statement. Where is evidence?

1.7 “As illustrated in Fig. 2, the visualization shows the impact of the proposed methods for subsequent tasks”: Figure 2 only presents heatmaps for individual images from the baseline and MIN, which does not adequately demonstrate "the impact of the proposed methods on subsequent tasks."


2. Methodological complexity and clarity issues. The training procedure involves multiple components (analytic classifier updates, auxiliary classifier, noise generators), which are hard to follow. For example, the role of the auxiliary classifier $W_{aux}$ in Eq.(14) and the sequence of updates (train $W_t$, then $P_t$, then $W_t$ again) are dense and confusing. Clear details of how the noise weights $\omega_i$ are learned (the prototype similarity in Eq. (11)-(12) is not fully justified) would be necessary. Overall, some parts of Section 4.2 and the training pipeline need to be more clearly presented.

3. Experimental clarity. Some details are either missing or need further clarification. It is not specified whether the results were averaged over random seeds for different task orderings, nor are standard deviations reported. In many cases, the performance improvement is minor, and without statistics such as standard deviation, it is not justified to label it as an improvement. In class incremental scenarios, the order of tasks will affect the performance which should be considered in experiments. Additionally, it is unclear whether all baseline methods were re-implemented under identical conditions (e.g., the same backbone, training schedule, and data preprocessing).

4. Ablation studies: The ablation study is not convincing in demonstrating the effectiveness of the proposed method. From Table 4, the improvement with and without Noise Expansion is minimal (less than 1% in some cases). Without standard deviation and using different task orderings, the results are not valid. No ablation study provided for hyperparameters.

**Rationales Behind Reviewer's Ratings:**

**Quality (3: Good):** The proposed method appears to be technically sound, and the experiments are comprehensive. However, there is limited theoretical analysis, and some methodological aspects (e.g. the prototype-based mixing) are poorly justified.

**Clarity (2: Fair):** The paper is not well-written. Many sections (notably Introduction, Section 4.2 and the algorithmic details) are very unclear and dense.

**Significance (2: Fair):** The performance improvement over prior methods is incremental (few percent gain on benchmarks). While the idea of beneficial noise is interesting, it is not significantly novel.

**Originality (3: Good):** The concept of injecting learned noise has appeared in other contexts (c.f. Pi-Noise literature), and using an auxiliary classifier for incremental learning has precedents. The specific combination (Noise Expansion + Mixture) is a novel twist, but it is not a fundamentally new paradigm.

---

> ### Author Rebuttal · Authors · 2025-07-30
>
> We sincerely thank you for the positive feedback on our **novel** recasting of forgetting, the **comprehensive experiments**, and the **parameter efficiency**. Please find our responses to the comments as follows:
>
> ## Reply to W1 & Q1 & Q2
>
> >**W1 & Q1 & Q2: Unclear statements in the introduction**:
>
> **Reply**: Thanks for pointing out some unclear expressions in Introduction. Due to limitations of rebuttal rules, we have to exclude the detail response of this part. We promise that we will polish the inexact statements in the revision. We can also reply these questions during the discussion phase.
>
> ## Reply to W2
>
> >**W2: Methodological complexity and clarity issues.**
>
> **Reply:** Thanks  for your suggestions about improving the clarity of Sec. 4.2. We provide the pseudo code of training pipeline in **Sec. 5 of supplementary**.
>
> According to your suggestion, we intend to **revise the Sec. 4.2 to extend more details about the optimization process**, such as visualizations of learned weight $\omega$ across the incremental phase.
>
> ## Reply to W3 & Q4
>
> >W3 & Q4:  Experimental details
>
> **Reply**:
>
> 1. (Different task orderings) In Sec. 5.1, line 205-206, we state that the learning order is determined by the **random seed 1993** and all methods are **run 3 times** to report the average results. **This setting is the same as most of previous works (MOS, EASE, APER, and RanPAC).**
>
> 2. (STD) We run baselines 3 times and report the average results. The results with stds are shown in Tab. 1, 2 and 3. We did not report the stds in the main paper **limited by the table width**. And most previous works also did not report the stds (MOS, EASE, ...).
>
> *Tab. 1*
>
> ||CIFAR||||||CUB||||||
> |-|:-:|:-:|:-:|:-:|:-:|:-:|:-:|:-:|:-:|:-:|:-:|:-:|
> ||T=10||T=20||T=50||T=10||T=20||T=50||
> |Methods|Avg.|Last|Avg.|Last|Avg.|Last|Avg.|Last|Avg.|Last|Avg.|Last|
> |L2P|±0.78|±0.57|±0.54|±0.33|±0.41|±0.26|±0.32|±0.36|±0.18|±0.25|±0.47|±0.32|
> |DualPrompt|±0.55|±0.79|±0.62|±0.85|±0.35|±0.41|±0.54|±0.75|±0.65|±1.05|±0.45|±0.82|
> |CODAPrompt|±0.32|±0.87|±0.70|±0.25|±0.58|±0.48|±0.87|±1.05|±0.37|±0.55|±0.25|±0.32|
> |ACIL|±0.03|±0.05|±0.03|±0.02|±0.07|±0.15|±0.21|±0.16|±0.18|±0.17|±0.33|±0.19|
> |SLCA|±0.89|±0.44|±0.76|±0.55|±0.30|±0.57|±0.44|±0.57|±0.34|±0.28|±0.47|±0.62|
> |FeCAM|±0.15|±0.17|±0.11|±0.26|±0.17|±0.39|±0.27|±0.33|±0.25|±0.20|±0.34|±0.43|
> |RanPAC|±0.07|±0.11|±0.15|±0.12|±0.10|±0.08|±0.14|±0.15|±0.17|±0.20|-|-|
> |APER|±0.05|±0.14|±0.09|±0.11|±0.09|±0.08|±0.15|±0.10|±0.08|±0.05|±0.13|±0.15|
> |EASE|±0.27|±0.88|±0.33|±0.52|±0.25|±0.68|±0.87|±1.02|±0.65|±0.62|±0.44|±0.32|
> |COFiMA|±0.16|±0.08|±0.27|±0.11|–|–|±0.35|±0.88|±1.22|±1.05|-|-|
> |MOS|±0.12|±0.21|±0.06|±0.25|±0.04|±0.18|±0.30|±0.46|±0.35|±0.18|±0.12|±0.20|
> |**MIN**|**±0.05**|**±0.16**|**±0.04**|**±0.29**|**±0.20**|**±0.36**|**±0.15**|**±0.18**|**±0.12**|**±0.14**|**±0.14**|**±0.22**|
>
> *Tab. 2*
>
> ||INA||||||INR||||||
> |-|:-:|:-:|:-:|:-:|:-:|:-:|:-:|:-:|:-:|:-:|:-:|:-:|
> ||T=10||T=20||T=50||T=10||T=20||T=50||
> |Methods|Avg.|Last|Avg.|Last|Avg.|Last|Avg.|Last|Avg.|Last|Avg.|Last|
> |L2P|±1.22|±1.06|±1.35|±2.08|±0.88|±0.62|±0.36|±0.56|±0.36|±0.45|±0.22|±0.18|
> |DualPrompt|±0.85|±0.65|±1.02|±0.84|±0.77|±1.06|±0.25|±0.65|±0.33|±0.40|±0.28|±0.22|
> |CODAPrompt|±1.35|±1.27|±2.08|±1.55|±0.66|±0.82|±0.59|±0.47|±0.56|±0.51|±0.36|±0.12|
> |ACIL|±0.42|±0.66|±0.57|±0.55|±0.36|±0.21|±0.05|±0.22|±0.08|±0.16|±0.09|±0.16|
> |SLCA|-|-|-|-|-|-|±0.26|±0.34|±0.22|±0.28|±0.31|±0.18|
> |FeCAM|±1.05|±0.65|±1.15|±0.94|±0.85|±0.65|±0.22|±0.18|±0.15|±0.26|±0.17|±0.11|
> |RanPAC|±1.35|±0.45|±1.07|±0.65|-|-|±0.31|±0.16|±0.09|±0.15|±0.27|±0.16|
> |APER|±0.72|±0.66|±1.05|±0.36|±0.82|±0.26|±0.11|±0.04|±0.15|±0.02|±0.10|±0.04|
> |EASE|±1.15|±1.34|±0.98|±1.33|±0.78|±0.92|±0.32|±0.19|±0.20|±0.09|±0.31|±0.24|
> |COFiMA|±0.86|±0.78|±0.75|±1.04|-|-|±0.15|±0.12|±0.16|±0.14|-|-|
> |MOS|±0.62|±0.92|±0.44|±0.32|±0.65|±0.45|±0.34|±0.20|±0.22|±0.12|±0.15|±0.08|
> |**MIN**|**±0.45**|**±0.89**|**±0.60**|**±0.26**|**±0.35**|**±0.27**|**±0.19**|**±0.07**|**±0.17**|**±0.15**|**±0.16**|**±0.19**|
>
> *Tab. 3*
>
> ||Food||||||Omni.||||||
> |-|:-:|:-:|:-:|:-:|:-:|:-:|:-:|:-:|:-:|:-:|:-:|:-:|
> ||T=10||T=20||T=50||T=10||T=20||T=50||
> |Methods|Avg.|Last|Avg.|Last|Avg.|Last|Avg.|Last|Avg.|Last|Avg.|Last|
> |L2P|±0.65|±0.85|±0.54|±0.52|±0.38|±0.26|±0.33|±0.26|±0.34|±0.16|±0.47|±0.22|
> |DualPrompt|±0.45|±0.40|±0.37|±0.26|±0.58|±0.67|±0.18|±0.45|±0.35|±0.27|±0.56|±0.38|
> |CODAPrompt|±0.33|±0.15|±0.30|±0.26|±1.77|±0.57|±0.42|±0.65|±0.18|±0.22|±0.14|±0.30|
> |ACIL|±0.03|±0.07|±0.08|±0.07|±0.04|±0.05|±0.35|±0.52|±0.26|±0.47|±0.18|±0.44|
> |SLCA|±0.37|±0.17|±0.27|±0.08|±0.16|±0.12|±0.26|±0.19|±0.44|±0.75|±0.21|±0.15|
> |FeCAM|±0.50|±0.42|±0.32|±0.31|±0.24|±0.11|±0.16|±0.22|±0.18|±0.16|±0.26|±0.31|
> |RanPAC|±0.32|±0.15|±0.20|±0.15|±0.16|±0.09|±0.12|±0.18|±0.08|±0.05|±0.10|±0.05|
> |APER|±0.19|±0.05|±0.18|±0.05|±0.08|±0.04|±0.20|±0.14|±0.21|±0.16|±0.08|±0.10|
> |EASE|±0.25|±0.22|±0.28|±0.35|±0.30|±0.33|±0.45|±0.40|±0.37|±0.25|±0.48|±0.37|
> |COFiMA|±0.06|±0.08|±0.15|±0.16|-|-|±0.40|±0.28|±0.42|±0.25|-|-|
> |MOS|±0.11|±0.08|±0.18|±0.22|±0.10|±0.08|±0.11|±0.15|±0.08|±0.12|±0.12|±0.15|
> |**MIN**|**±0.07**|**±0.06**|**±0.12**|**±0.16**|**±0.07**|**±0.12**|**±0.04**|**±0.10**|**±0.03**|**±0.11**|**±0.15**|**±0.19**|
>
> 3. (Different learning orders) We perform the experiments of **different random seeds** {1993, 1994, 1995, 1996, 1997, 1998} in ImageNet-R 10 steps setting, and report the results in **Sec. 2 of the supplementary**. The results are shown in Tab. 4.
>
> *Tab. 4*
>
> |Methods|ViT-B/16-IN21K||ViT-B/16-IN1K||
> |:-|:-:|:-:|:-:|:-:|
> ||Avg.|Last|Avg.|Last|
> |SLCA|81.69±1.50|76.09±0.52|81.97±0.98|76.62±1.12|
> |FeCAM|78.23±0.54|72.11±1.06|78.43±1.06|72.49±0.74|
> |RanPAC|81.29±0.73|76.44±0.93|81.75±0.92|76.99±0.67|
> |APER|74.98±0.70|67.56±1.06|73.41±1.00|66.71±0.61|
> |EASE|81.49±0.35|75.79±0.65|81.84±0.59|76.28±0.31|
> |COFiMA|81.16±1.71|75.46±0.86|82.04±0.97|76.54±1.61|
> |MOS|82.16±0.27|77.55±0.36|80.66±0.59|74.21±0.16|
> |**MIN**|**85.00±0.28**|**79.52±0.35**|**85.66±0.59**|**80.27±0.52**|
>
> 4. (Conditions) We ensure all baselines are compared under fair conditions. First, all baselines adopt **the same backbones** (ViT-B/16-IN21K from timm library and ViT-B/16-IN1K downloading from Huggingface) and **the exemplar-free setting**. Second, they are trained with **the same learning rate, weight decay, optimizer and scheduler**. Third, their own hyper-parameters are kept default. Therefore, their performance may have **slight differences** from the original paper.
>
> 5. (MOS for CUB) MOS provides no results under 10 tasks setting. It just reports the results of CUB B0-Inc10, equivalent to the 20 steps setting in our paper since CUB contains 200 categories. Their reported result (93.49) is closed to the one we reported (92.62). In addition, most of reported results are close with the original papers, and **in some datasets (Omni.) even higher than the original papers**.
>
> ## Reply to W4 & Q3 & Q5
>
> >**W4 & Q3 & Q5: More ablation studies and sensitivity**:
>
> **Reply**:
>
> 1. (STD) **Limited by the table width**, we did not report the standard deviations in the main text. The results of the ablation study with  the standard deviations are reported in the following table (Tab. 5). Compare row 3, 4 and 9, the results show that the contributions of each component are stable and effective.
>
> *Tab. 5*
>
> ||CIFAR||CUB||IN-A||IN-R||Food||Omni||
> |:-|:-:|:-:|:-:|:-:|:-:|:-:|:-:|:-:|:-:|:-:|:-:|:-:|
> |Methods|Avg.|Last|Avg.|Last|Avg.|Last|Avg.|Last|Avg.|Last|Avg.|Last|
> |Baseline|91.18±0.03|86.82±0.07|91.55±0.16|87.08±0.17|62.23±0.45|53.65±0.57|75.14±0.05|68.45±0.27|90.26±0.03|85.57±0.10|84.55±0.33|76.82±0.44|
> |NE+AvgN|94.07±0.04|90.22±0.10|92.89±0.09|90.77±0.15|71.43±0.55|62.59±0.70|83.86±0.24|78.65±0.11|92.67±0.05|89.34±0.09|86.77±0.15|80.12±0.18|
> |NE+OnlyMu|89.27±0.24|85.63±0.33|88.97±0.46|85.45±0.45|65.87±1.27|55.36±1.02|76.27±0.44|67.15±0.34|88.45±0.09|83.16±0.12|82.15±0.68|73.95±0.70|
> |NE+OnlySigmma|94.37±0.05|91.32±0.12|93.15±0.17|90.47±0.09|70.45±0.25|61.32±0.33|84.15±0.20|79.05±0.12|92.90±0.05|89.25±0.04|87.06±0.07|80.12±0.05|
> |NE+LastN|92.36±0.18|87.92±0.27|92.37±0.08|87.25±0.12|63.23±0.75|54.15±0.92|79.15±0.16|71.71±0.28|90.62±0.12|86.12±0.05|85.02±0.06|77.47±0.08|
> |NE+RandomSN|92.65±0.21|88.10±0.15|92.15±0.23|88.76±0.26|64.39±1.08|54.88±1.35|81.39±0.65|73.15±0.81|91.08±0.30|86.57±0.19|84.77±0.24|77.96±0.16|
> |**MIN**|**95.12±0.05**|**92.12±0.16**|**94.00±0.15**|**91.22±0.18**|**72.89±0.45**|**64.32±0.89**|**85.18±0.19**|**79.75±0.07**|**93.36±0.07**|**90.04±0.06**|**87.36±0.04**|**80.55±0.10**|
>
> 2. (Sensitivity of $d_2$) We have performed the experiment in sensitivity of the dimensionality $d_2$, and provide the results in supplementary materials (**Sec. 4 in the supplementary**). The results show that it achieves the best trade off between performance and the number of parameters when setting the ${d_2}=192$.
> 3. (Sensitivity of $\tau$) We keep the temperature ${\tau}=2$ in all experiments in the main text. According to your suggestion, we perform the sensitivity experiment on the temperature $\tau$. The results are shown in the following table (Tab. 6). The results indicate that the model performance is not sensitive to the hyper-parameter $\tau$.
>
> *Tab. 6*
> |$\tau$=0.50||0.75||1.50||2.00||
> |:-:|:-:|:-:|:-:|:-:|:-:|:-:|:-:|
> |Avg.|Last|Avg.|Last|Avg.|Last|Avg.|Last|
> |84.92±0.21|79.30±0.23|84.79±0.22|79.38±0.10|84.91±0.19|79.59±0.12|85.18±0.19|79.75±0.07|
>
> 4. (More detailed ablation studies) We have explored more ablation studies with the *Noise Mixture* in Tab. 5.
>
>    In Tab. 2, **AvgN** denotes the method with a simple average of noises, which is the same as the row 4 of Tab. 4 in the main text, **LastN** denotes the method using only the noise from last noise and **RandomSN** denotes the method with generating noise by selecting a task randomly.
>
>    A comparison of experimental results for **AvgN**, **LastN**, and **RandomSN** demonstrates that **integrating noise injection across multiple tasks is superior to single-task noise injection alone**.

---

> > ### Comment · Reviewer_wYxB · 2025-08-04
> >
> > Thank you very much for the rebuttal. However, after carefully reviewing the authors' responses and thoroughly reconsidering the manuscript, I maintain my original score. The rebuttal, while addressing several minor concerns adequately, does not sufficiently resolve the key limitations initially highlighted.
> >
> > Specifically, the major concerns remain as follows:
> >
> > 1. Fundamental Conceptual Issues: Fundamental conceptual issues related to "non-essential features," "task-specific redundancy," and "parameter sensitivity" remain unaddressed. These terms are central motivations for the proposed method, and without precise definitions or clear theoretical justification, the paper's foundational basis remains unclear.
> >
> > 2. Methodological Complexity: While the authors propose adding pseudo-code to address the complexity concerns, the methodological clarity issues raised such as the precise role and necessity of multiple training steps and auxiliary classifiers remain unresolved.
> >
> > 3. Missing Dataset Information
> > Table 6 (temperature sensitivity) provides results but doesn't specify which dataset and setting these results are from, making it impossible to verify consistency with other reported numbers.
> >
> > 4. Novelty Concerns: While the approach combining noise injection and mixture weights introduces an interesting angle, the novelty remains limited. The core idea of beneficial noise and auxiliary classifiers (i.e., task-specific adapters with learned weighted mixing) already exists in related literature, making this work more of an incremental refinement than a fundamentally novel contribution.
> >
> > In summary, the authors' responses address peripheral concerns effectively but fail to alleviate significant issues related to fundamental conceptual clarity, methodological complexity, incremental significance, and limited novelty.

---

> > > ### Author Response · Authors · 2025-08-05
> > >
> > > We appreciate your feedback. **Due to the length constraints of the rebuttal letter**, we could not fully reply all concerns at previous phase. Below we have provided detailed supplementary responses：
> > >
> > > ## Reply to Fundamental Conceptual Issues
> > >
> > >  1. "non-essential features": We use this concept to refer to the features that should not be included within the decision boundary.
> > >
> > >     For example, even if we **freeze the pre-trained backbone and only train a FC layer incrementally** on several downstream tasks, the model can only recognize the classes from the last task. But if we adopt the prototypes to use the distance measurement (e.g. cosine similarity) for classification, pre-trained model can **work for all tasks** (This phenomenon has been mentioned in many related studies such as Aper).
> > >
> > >     This situation can be certainly attributed to the parameter drift or bias of the classifier but these reasons are general and easily confusing with other phenomenons such as representational overlap due to parameter drift of the backbone network.
> > >
> > >     In the described phenomenon, the representations output by the pretrained model remain unchanged, while only the decision boundary learned by the classifier shifts. This kind of catastrophic forgetting demonstrates that **the classifier integrates features from the pre-trained model that are irrelevant to the current task into its decision boundary**.
> > >
> > > 2. "task-specific redundancy": We use this concept to refer to the **over-parameterization** of pre-trained models, i.e., the model size of pre-trained is much larger than scale of downstream sub-tasks. In this context of over-parameterization, **models naturally exhibit redundancy for individual tasks**.
> > >
> > > 3. "parameter sensitivity": We employ this concept to demonstrate that PTMs tend to **overfit on individual tasks**, converging rapidly on each downstream task. When developing a model that excels across multiple tasks, this overfit can readily cause task confusion.
> > >
> > >
> > >
> > > ## Reply to Methodological Complexity
> > >
> > >  1. "Necessity  of auxiliary classifiers": The analytic classifier employed in our study determines the weights through an iterative linear regression approach, as detailed in Sec. 3 of the main text, which means **this classifier does not employ back-propagation for gradient updates**.
> > >
> > >     Consequently, **an additional FC layer must be utilized as an auxiliary classifier to train the Pi-Noise layer**, enabling proper gradient back-propagation.
> > >
> > >     We intend to **incorporate the explanation for employing auxiliary classifiers within the main text**.
> > >
> > > 2. "Necessity of multiple training steps": The training pipeline of each incremental stage is divided into **3 phases**.
> > >
> > >    First, we update the analytic classifier through an iterative linear regression approach.
> > >
> > >    Second, we initiate the auxiliary classifier with the weight of analytic classifier and update both the Pi-noise layers and auxiliary classifier through back-propagation.
> > >
> > >    In the end, we update the analytic classifier through an iterative linear regression approach to integrate the new task information learned from the second phase into the auxiliary classifier.
> > >
> > >
> > >
> > > ## Reply to Missing Dataset Information Table 6
> > >
> > > We apologize for missing dataset information of Tab. 6 since we delete all table information due to the length constraints of the rebuttal letter. **This experiment performs in the  setting of ImageNet-R 10 steps.** We run each experiment 3 times and report the average performance and corresponding std information.
> > >
> > >
> > >
> > > ## Reply to Novelty Concerns
> > >
> > > To the best of our knowledge, we are **the first to apply beneficial noise in the CIL field** and propose the first beneficial noise-based solution.
> > >
> > > Furthermore, there appear to be no existing studies addressing CIL from the perspective of noise. Task-specific adapters serve merely as an implementation approach in our work, rather than constituting the core contribution.
> > >
> > > Moreover, **other reviewers have acknowledged the insight and novelty of our approach**. We contend that this innovative perspective on noise represents merely a starting point, potentially enabling the CIL community to develop more diverse and enriched solutions.

---

> > > ### Author Response · Authors · 2025-08-06
> > >
> > > Dear Reviewer,
> > >
> > > Thank you sincerely for your response to our previous rebuttal.
> > >
> > > I wanted to follow up to ensure that we have adequately addressed all of your concerns. Could you please let us know if our explanations and revisions have satisfactorily resolved the issues you raised? If there are any remaining concerns or if you need further clarification on any points, we would be happy to provide additional information.
> > >
> > > We appreciate your time and valuable feedback throughout this review process.
> > >
> > > Best regards,
> > >
> > > All of Authors.

---

### Official Review · Reviewer_xsXt · 2025-06-26

**Clarity:** 3
**Significance:** 3
**Originality:** 3
**Rating:** 4
**Confidence:** 4

**Summary:**

This paper focuses on class incremental learning and proposes a new noise perspective to rethink the parameter drift problems in this task. Based on this perspective, a noise mix method was proposed to integrate the incremental knowledge and previous knowledge. Various experiments show that this perspective and method are effective.

**Questions:**

Can your method be extended to the incremental learning in a generative model, for instance, to learn some new concepts to generate, but without destroying the previous concept generation?

**Ethical Concerns:**

["NO or VERY MINOR ethics concerns only"]

**Final Justification:**

My main concern about the noise explanation is partly resolved. Due to the policy limitation, the authors can not provide the visualization results in the rebuttal, but promise to supplement the visualization results in a future version. I decided to keep my original score.

**Limitations:**

yes

**Paper Formatting Concerns:**

N.A.

**Quality:**

3

**Strengths And Weaknesses:**

Pros:
1) This paper provides a new noise perspective regarding the parameter drift or forgetting problem, very interesting.
2) Based on this kind of noise perspective, this paper designs some filters or integrating mechanisms to maintain the good noise and remove others.
3) This perspective and framework are general and can be easily extended to other semi-dense (like detection) or dense prediction tasks.
4) The results show consistent improvements, and the whole experiment with ablations is sufficient.
5) The whole writing is very clear and smooth, organized well, and easy to understand.

Cons:
You should add some visualization or other high-level intuition explanations to make readers accept the noise perspective more easily and naturally.

---

> ### Author Rebuttal · Authors · 2025-07-30
>
> We sincerely thank you for highlighting the **novelty** of our noise perspective on parameter drift, the **effectiveness** of our designed filters, the framework's **generalizability** to other tasks, the **consistent improvements** and the **sufficient** ablation study, and the **clear writing**. Please find our responses to the comments as follows:
>
> >W1: Add some visualization or other high-level intuition explanations to make readers accept the noise perspective more easily and naturally.
>
> **Reply to W1**: Thanks for your insightful suggestion. Due to the limitation of rebuttal rules, we cannot upload any figures. We intend to add the visualization of intermediate features before and after noise injection to intuitively demonstrate the impact of the beneficial noise on these features. Furthermore, heat-map visualizations of learnable weights $w$ will be added to intuitively illustrate the dynamics of noise learning.
>
> >Q1: Can your method be extended to the incremental learning in a generative model, for instance, to learn some new concepts to generate, but without destroying the previous concept generation？
>
> **Reply to Q1**:  It is interesting and promising to extend the MIN to generative model. We find highly related paper [1]. After reading the paper, we believe that MIN could potentially extend to generative incremental learning as its core mechanism.
>
> By adding Pi-Noise layers to U-Net/DiT, it avoids destructive parameter drift in learning the new concepts to generate. This aligns with generative models' need to preserve core distributional features.
>
> Its ability to suppress irrelevant activations indicates that it could also help generative models maintain fidelity to prior concepts. By masking inefficient patterns in intermediate features, it may prevent new concepts from overwriting existing latent representations.
>
> [1] Continual learning of diffusion models with generative distillation, 3rd Conference on Lifelong Learning Agents (CoLLAs), 2024.

---

> > ### Comment · Reviewer_xsXt · 2025-08-05
> >
> > My main concern about the noise explanation is partly resolved. Due to the policy limitation, the authors can not provide the visualization results in the rebuttal, but promise to supplement the visualization results in a future version. I decided to keep my original score.

---

### Official Review · Reviewer_Cck3 · 2025-06-28

**Clarity:** 2
**Significance:** 2
**Originality:** 2
**Rating:** 4
**Confidence:** 4

**Summary:**

This manuscript focused on the class incremental learning (CIL) with pre-trained models (PTM). Based on the analytic learning methods [1], this work considered leveraging the beneficial noise and proposed Mixture of Noise (MIN). Specifically, a task-specific adapter is based on the task-specific noise learned from each task. Then, a set of dynamic weights was used to adjust the optimal mixture of different task noises. Based on this design, the beneficial noise was embedded into the intermediate features to mask the response of inefficient patterns. Experiments on some commonly used benchmarks were conducted to demonstrate the effectiveness of the proposed method.

**Questions:**

- 1. I noticed that the reparameterization process was introduced within the adapter $P_t$. Then, I have two questions regarding this process:
    - 1.1 During the inference process, do we need to sample $\epsilon$ multiple times for each sample $x$? If not, please explain the reason.
    - 1.2 If so, how do you determine the network output? Based on my understanding, for each input $x$, we need to sample $\epsilon$ for multiple times $\epsilon^{(1)}, \epsilon^{(2)}, \cdots, \epsilon^{(k)}$, we will also get multiple $\epsilon_{t}$ as  $\epsilon^{(1)}_{t}, \epsilon^{(2)}_{t}, \cdots, \epsilon^{(k)}_{t}$  for  $k$ samplings. How should we compute the weighting process in Eq.(13) and the final output of the neural network?

- 2. In Line 141, the authors claimed that the positive noise can be learned to "suppress the confusing pattern across tasks through a large interference to mask irrelevant activation in the features". I guess the irrelevance and relevance were measured by the cosine similarity between the reparameterized noise center $\mu$. Based on this consideration, how about directly adopting a single parameter $\mu$ to make it a deterministic adapter?

- 3. About the motivation behind the auxiliary classifier. What is the objective or consideration for introducing this auxiliary classifier? I didn't get it.

- 4. Some discussions regarding the high-level ideas of the proposed method. When I first read the methodological parts, especially Sections 4.1 and 4.2, I naturally connected them with some existing concepts, including "Bayesian Neural Network" and "uncertainty" which have been widely investigated in deep learning. This manuscript explained the proposed method under the concept of positive noise and ignored some descriptions about the relevant details, e.g., the multiple sampling regarding $\epsilon_{1}, \epsilon_{2}, \cdots, \epsilon_{t}$. This made some steps confusing. Could the author discuss the relationship between the proposed method and the studies about "Bayesian Neural Network" or "uncertainty"?

**Ethical Concerns:**

["NO or VERY MINOR ethics concerns only"]

**Final Justification:**

During the rebuttal period, my questions have been well answered. I tend to maintain my positive rating.

**Quality:**

3

**Strengths And Weaknesses:**

## Strengths
1. This manuscript adopted a special perspective, the beneficial noise, to design the task-specific adapters, which is novel to the reviewer. Although it was still under the framework of parameter-efficient adapters, it brought a new perspective to designing the adapters.
2. The experimental part is complete. Most recent baseline methods and commonly used datasets have been included. The ablation study is clear.
3. Source code was provided to help the readers verify the reproducibility.


## Weaknesses
1. Although this manuscript tried to use "beneficial noise" to "mask the irrelevant activation in the features", the explanation of this design was still not clear for me. Based on my understanding, the MIN module seems like a "Bayesian Parameter-Efficient Adapter" since it introduces some uncertainties through noisy sampling within the parameters of the intermediate layer. However, I didn't see any discussion related to the "Bayesian" or "uncertainty". Also, the processing of the network output (since it seems that we have an extra sampling process for $\epsilon$) is not clear.
2. The motivation of the proposed method is still confusing. The connection between the methodology and some claims was not straightforward. The authors introduced some expressions, like "learn positive noise to suppress the confusing pattern across tasks" in Line 141, but I didn't find any connections with the design. See the Questions part for more details.
3. Some methodological steps were not clear to the reader. For example, the motivation for introducing the auxiliary classifier in Eq. (14) is not clear.

---

> ### Author Rebuttal · Authors · 2025-07-30
>
> We sincerely thank you for recognizing the **novelty** of our beneficial noise perspective, the **completeness of experiments** with recent baselines and ablation studies, and the provision of source code for **reproducibility**. Please find our responses to the comments as follow:
>
> ## Reply to W1 & Q4
>
> >W1 & Q4: Strengthen the distinctions and connection with existing theories, specifically Bayesian networks and uncertainty modeling.
>
> **Reply**: Thank you for your insightful observation regarding connections to Bayesian Neural Networks (BNNs) and uncertainty modeling. We acknowledge that the introduction of noise and its generation might partly resemble some aspects of BNNs.
>
> However, it is crucial to clarify that our MIN framework fundamentally differs in its objective and mechanism from traditional BNNs or uncertainty modeling. **MIN adopts noise layer to construct a buffer of confusing feature rather than quantifying prediction uncertainty**. The noise generators learn to mask cross-task interference patterns, analogous to targeting "confusion uncertainty" in feature space.
>
> Specifically, although they are similar with each other (from the aspect of terminology and mechanism), the main differences are:
>
> 1. The primary function of this noise in MIN is direct feature manipulation for CIL, not uncertainty estimation.
> 2. While we sample noise from a standard normal distribution, this serves primarily as a reparameterization trick to enable backpropagation for training the noise generators. This approach is distinct from Bayesian methods, which aim to model posterior distributions to quantify epistemic and aleatoric uncertainties in parameters or predictions.
>
> ## Reply to Q1
>
> >Q1: Does the inference require multiple samplings? Explain the reason.
>
> **Reply**: A Pi-Noise layer is integrated into each layer of the model. During inference, each layer samples $\epsilon$ **once** and injects it into the features of that layer. Therefore, the model actually **samples multiple $\epsilon$ during each inference pass, yet only needs to operate on a single $\epsilon$ per layer**.
>
> ### **The forward propagation during training:**
>
> During the initial stages of training, models may indeed produce uncertain outcomes due to the randomness of injected noise. This uncertainty is precisely what building a robust classifier requires.
>
> Specifically, a zero-mean noise sampled from $P_t$ parameterized by $\sigma$ is injected into input features. After supervised training, representations highly relevant to the current task acquire smaller $\sigma$ values to mitigate predictive uncertainty, whereas less relevant representations adopt larger $\sigma$ values.
>
> Building upon this, ${\cal F}$ in Eq. (1) exhibits higher uncertainty toward low-correlation representations. Consequently, when calculating weights via Eq. (3), **it reduces the weighting of representations with low correlations**, thereby decreasing reliance on unrelated features. Therefore, the updated classifier exhibits robustness to low-correlation representations, and **its output remains stable even without multiple sampling**.
>
> ### **The forward propagation after training (during inference):**
>
> The main reasons why we just sample once are:
>
> 1. After well training MIN, the output remains stable across multiple independent noise sampling. And additional overload is introduced by multiple sampling but it does not improve the performance.
> 2. From the additional results in Tab. 1, the std of performance is rather small, which shows the stability without multiple sampling.
>
> ## Reply to W2 & Q2
>
> >W2 & Q2:  How about directly adopting a single parameter $\mu$ to make it a deterministic adapter? Clarify the connection with the design.
>
> **Reply**: Thanks for your suggestion. We have explored more ablation studies in the following table (Tab. 1).
>
> *Tab. 1 Results of ablation study.*
>
> |               |     CIFAR      |                |      CUB       |                |      IN-A      |                |      IN-R      |                |      Food      |                |      Omni      |                |
> | :------------ | :------------: | :------------: | :------------: | :------------: | :------------: | :------------: | :------------: | :------------: | :------------: | :------------: | :------------: | :------------: |
> | Methods       |      Avg.      |      Last      |      Avg.      |      Last      |      Avg.      |      Last      |      Avg.      |      Last      |      Avg.      |      Last      |      Avg.      |      Last      |
> | Baseline      |   91.18±0.03   |   86.82±0.07   |   91.55±0.16   |   87.08±0.17   |   62.23±0.45   |   53.65±0.57   |   75.14±0.05   |   68.45±0.27   |   90.26±0.03   |   85.57±0.10   |   84.55±0.33   |   76.82±0.44   |
> | NE+AvgN       |   94.07±0.04   |   90.22±0.10   |   92.89±0.09   |   90.77±0.15   |   71.43±0.55   |   62.59±0.70   |   83.86±0.24   |   78.65±0.11   |   92.67±0.05   |   89.34±0.09   |   86.77±0.15   |   80.12±0.18   |
> | NE+OnlyMu     |   89.27±0.24   |   85.63±0.33   |   88.97±0.46   |   85.45±0.45   |   65.87±1.27   |   55.36±1.02   |   76.27±0.44   |   67.15±0.34   |   88.45±0.09   |   83.16±0.12   |   82.15±0.68   |   73.95±0.70   |
> | NE+OnlySigmma |   94.37±0.05   |   91.32±0.12   |   93.15±0.17   |   90.47±0.09   |   70.45±0.25   |   61.32±0.33   |   84.15±0.20   |   79.05±0.12   |   92.90±0.05   |   89.25±0.04   |   87.06±0.07   |   80.12±0.05   |
> | NE+LastN      |   92.36±0.18   |   87.92±0.27   |   92.37±0.08   |   87.25±0.12   |   63.23±0.75   |   54.15±0.92   |   79.15±0.16   |   71.71±0.28   |   90.62±0.12   |   86.12±0.05   |   85.02±0.06   |   77.47±0.08   |
> | NE+RandomSN   |   92.65±0.21   |   88.10±0.15   |   92.15±0.23   |   88.76±0.26   |   64.39±1.08   |   54.88±1.35   |   81.39±0.65   |   73.15±0.81   |   91.08±0.30   |   86.57±0.19   |   84.77±0.24   |   77.96±0.16   |
> | **NE+NM**     | **95.12±0.05** | **92.12±0.16** | **94.00±0.15** | **91.22±0.18** | **72.89±0.45** | **64.32±0.89** | **85.18±0.19** | **79.75±0.07** | **93.36±0.07** | **90.04±0.06** | **87.36±0.04** | **80.55±0.10** |
>
> **OnlyMu** denotes the method adopting **a single parameter $\mu$**. However, the strategy of solely using $\mu$ demonstrates **adverse effects** in the most of datasets. Furthermore, we explore to **adopt a single parameter $\sigma$** to generate a series of zero-mean noise. The experimental results (**OnlySigmma**) shows excellent performance.
>
> We speculate that MIN **mainly works by the uncertainties introduced by the $\sigma$**. And the noise center $\mu$ provides a simple bias to increase the difference between tasks.
>
> In the absence of $\sigma$, the classifier is influenced by numerous low-correlation representations. Directly adjusting the feature center using $\mu$ induces alterations in certain low-correlation representations that previous tasks rely on, resulting in performance degradation.
>
> Conversely, after simultaneous application of $\mu$ and $\sigma$, the classifier becomes insensitive to low-correlation representations. Under these conditions, feature center adjustment with $\mu$ can enhance inter-task differences.
>
> ## Reply to W3 & Q3
>
> >W3 & Q3: Clarify the motivation behind the auxiliary classifier.
>
> **Reply to W3 & Q3**: The analytic classifier employed in our study determines the weights through an iterative linear regression approach, as detailed in Sec. 3 of the main text, which means **this classifier does not employ back-propagation for gradient updates**.
>
> Consequently, **an additional FC layer must be utilized as an auxiliary classifier to train the Pi-Noise layer**, enabling proper gradient back-propagation.
>
> We intend to **incorporate the explanation for employing auxiliary classifiers within the main text**. Appreciate your valuable suggestion.

---

> > ### Comment · Reviewer_Cck3 · 2025-08-04
> > **Response to the author rebuttal**
> >
> > I appreciate the authors' effort in answering my questions. My concerns have been addressed.
> >
> > I also read the reviews from my colleagues and noticed that they have some remaining questions. Before the end of the internal discussion period, I tend to maintain my current positive rating.

---

> > > ### Author Response · Authors · 2025-08-05
> > >
> > > Thanks sincerly for your positive feedback and we will try our best to address the concerns for all reviewers. Thank you again for your reminder and reply.

---

### Official Review · Reviewer_MbJ6 · 2025-07-03

**Clarity:** 3
**Significance:** 3
**Originality:** 3
**Rating:** 4
**Confidence:** 4

**Summary:**

This paper introduces a new method called Mixture of Noise (MIN) to address catastrophic forgetting during class-incremental learning.
The key idea is to view the problem of parameter drift, where the model forgets old information while learning new things, as a form of destructive noise.
To combat this, the proposed MIN method learns a beneficial noise for each new task.
The method then uses a Noise Mixture strategy to dynamically combine the beneficial noises from all the tasks the model has learned.
Experiments on several benchmark datasets show the effectiveness of MIN.

**Questions:**

Please see the weaknesses above.
I encourage the authors to address my concerns, and I am with a open mind on raising my score.

**Ethical Concerns:**

["NO or VERY MINOR ethics concerns only"]

**Final Justification:**

Thanks to the authors' efforts to address my concern. I think most of my major concerns are successfully addressed.
Overall, I will raise my score to `borderline accept' because some of my concerns are addressed, and I encourage the author to give rigorous proof of Eq. 3.

**Limitations:**

Yes

**Quality:**

3

**Strengths And Weaknesses:**

Strengths:

1. This paper is easy to follow.
2. The experiments provided show overall high performance in the benchmarks.
3. Using noise to enhance CIL performance is something new to me, and I believe it will introduce new insight to the community.

Weaknesses:

1. The paper uses Analytic Learning as its baseline. However, as described in Fig. 1(c), the PTM is updated in each stage through the proposed noise mechanism. The solution in Eq. (3) is derived from Eq. (1) under the assumption that the feature extractor F(·) is frozen during all training phases. In the paper's setting, where F(·) is modified, does the solution presented in Eq. (3) still hold? This needs further clarification.

2. Another major concern is that the proposed noise expansion and noise mixture mechanisms are functionally similar to previous prompt-based or adapter-based PEFT methods. They all introduce additional parameters into the transformer to adapt to new tasks. For the current version of this paper, I am more inclined to see this as an improved adapter mechanism with limited technical contribution. I suggest the authors add more insightful discussion, using theory or experiments, to further elaborate on the advantages of their approach over previous adapter-based mechanisms, specifically for CIL.

3. I believe the set of learnable weights  $w$ shown in Eq. (12) is a crucial component for mitigating forgetting, yet the paper provides insufficient experimental analysis for it. I suggest the authors include more ablation studies, for instance, by replacing the weighted summation in Eq. (13) with a simple average of the noises. Furthermore, it would be insightful to analyze or visualize the final learned weights $w$ after incremental tasks are completed.

---

> ### Author Rebuttal · Authors · 2025-07-30
>
> We sincerely thank you for the positive feedback highlighting the **clarity** and ease of following our methodology, the **high performance** demonstrated in our experiments, and the **novel insights** presented in our work. Please find our responses to the comments as follow:
>
> ## Reply to W1
>
> > W1: If F(·) is modified, does the solution presented in Eq. (3) still hold?.
>
> **Reply**:  The methods such as finetune and PEFT indeed update the ${\cal F}$ and learn new representations for a new task, so Eq. (3) does not hold. Therefore, as shown in the row "Finetune" of the following table (Tab. 1), even with an analytic classifier, fine-tuning the backbone network on each task still **leads to severe catastrophic forgetting**.
>
> *Tab. 1 Incremental trends of replacing the Pi-Noise with the different PEFT approaches under the setting of ImageNet-R 10 steps.*
>
> | Methods     |   task0   |   task1   |   task2   |   task3   |   task4   |   task5   |   task6   |   task7   |   task8   |   task9   |   Avg.    |
> | :---------- | :-------: | :-------: | :-------: | :-------: | :-------: | :-------: | :-------: | :-------: | :-------: | :-------: | :-------: |
> | Finetune    |   93.61   |   78.73   |   75.34   |   68.00   |   61.91   |   56.37   |   48.23   |   47.39   |   43.62   |   40.15   |   61.24   |
> | VPT-Shallow |   93.47   |   84.36   |   80.15   |   71.78   |   68.42   |   65.24   |   63.77   |   58.67   |   60.17   |   58.36   |   70.44   |
> | VPT-Deep    |   94.08   |   80.45   |   75.80   |   70.07   |   65.35   |   60.80   |   57.26   |   54.33   |   48.29   |   42.75   |   64.92   |
> | Adapter     |   94.45   |   78.65   |   70.24   |   65.33   |   57.26   |   45.46   |   40.23   |   35.66   |   35.27   |   30.80   |   55.34   |
> | **MIN**     | **94.12** | **91.42** | **88.05** | **86.00** | **85.34** | **83.81** | **82.15** | **81.55** | **80.25** | **79.65** | **85.23** |
>
> Instead of **learning new representation** like finetune and PEFT methods, the proposed MIN introduces additional noise layer to capture the confusing features **while holding the original representations**. The noise layer can be regarded as **a kind of buffer of these confusing features**, so that the representations of current task maintain more stable. Formally, the noise layer can be viewed as converting $z$ to $z+\epsilon$.
>
> It essentially still leverages the representational power of the pretrained model. **This noise layer does not disrupt the representation of the previous task**, so Eq. (3) still holds.
>
> ## Reply to W2
>
> >W2: More insightful discussion to further elaborate on the advantages of MIN over previous adapter-based mechanisms, specifically for CIL.
>
> **Reply** :  Current PEFT methods primarily focus on performance in new tasks by learning new representations with minimal parameters. However, the knowledge acquired from new tasks is destructive to previous tasks in CIL.
>
> As shown in Tab. 1, after fine-tuning with three PEFT schemes separately for each task, all these methods (VPT-Shallow, VPT-Deep and Adapter) **result in catastrophic forgetting**. Furthermore, methods with greater learning capabilities (Adapter) exhibit a more rapid decline in performance.
>
> In contrast, MIN does not learn new representation, so it can avoid drifting. The main differences and advantages are:
>
> 1. **Different Motivation:** MIN aims to adapt to downstream tasks through suppressing irrelevant feature representations by **leveraging the powerful representational ability of PTMs** rather than significantly updating the representations. The core premise is that the visual patterns captured by existing pre-trained models sufficiently encompass most downstream tasks.
> 2. **Stability caused by noise layer:** The core of MIN is the noise layer and it can be regarded as **a kind of buffer of these confusing features**, so that the representations of current task maintain more stable. Formally, the noise layer can be viewed as converting $z$ to $z+\epsilon$.
>
> ## Reply to W3.1
>
> >W3.1: More ablation studies such as replacing the weighted summation in Eq. (13) with a simple average of the noises.
>
> **Reply**:  We appreciate your suggestion and have included additional ablation study results in Tab. 2.
>
> In addition to the existing NE and NM components, we implement other **five** distinct methods as the alternatives for NM (*Noise Mixture*). In Tab. 2, **AvgN** denotes the method with a simple average of noises, which is the same as the row 4 of Tab. 4 in the main text, **OnlyMu** denotes the method with generating the deterministic noise only by vector $\mu$, **OnlySigmma** denotes the method with generating the zero-mean noise only by vector $\sigma$, **LastN** denotes the method using only the noise from last noise and **RandomSN** denotes the method with generating noise by selecting a task randomly.
>
> A comparison of experimental results for AvgN, LastN, and RandomSN demonstrates that **integrating noise injection across multiple tasks is superior to single-task noise injection alone**.
>
> In addition, we also add the ablation study suggested by Reviewer Cck3. Comparison of the experimental results between OnlyMu and OnlySigma demonstrates that **the uncertainty introduced by the vector $\sigma$ is crucial for operation of MIN**. In contrast, the deterministic adapter constructed solely from the $\mu$ vector negatively impacts the original baseline.
>
> *Tab. 2 Results of more ablation studies.*
>
> |               |     CIFAR      |                |      CUB       |                |      IN-A      |                |      IN-R      |                |      Food      |                |      Omni      |                |
> | :------------ | :------------: | :------------: | :------------: | :------------: | :------------: | :------------: | :------------: | :------------: | :------------: | :------------: | :------------: | :------------: |
> | Methods       |      Avg.      |      Last      |      Avg.      |      Last      |      Avg.      |      Last      |      Avg.      |      Last      |      Avg.      |      Last      |      Avg.      |      Last      |
> | Baseline      |   91.18±0.03   |   86.82±0.07   |   91.55±0.16   |   87.08±0.17   |   62.23±0.45   |   53.65±0.57   |   75.14±0.05   |   68.45±0.27   |   90.26±0.03   |   85.57±0.10   |   84.55±0.33   |   76.82±0.44   |
> | NE+AvgN       |   94.07±0.04   |   90.22±0.10   |   92.89±0.09   |   90.77±0.15   |   71.43±0.55   |   62.59±0.70   |   83.86±0.24   |   78.65±0.11   |   92.67±0.05   |   89.34±0.09   |   86.77±0.15   |   80.12±0.18   |
> | NE+OnlyMu     |   89.27±0.24   |   85.63±0.33   |   88.97±0.46   |   85.45±0.45   |   65.87±1.27   |   55.36±1.02   |   76.27±0.44   |   67.15±0.34   |   88.45±0.09   |   83.16±0.12   |   82.15±0.68   |   73.95±0.70   |
> | NE+OnlySigmma |   94.37±0.05   |   91.32±0.12   |   93.15±0.17   |   90.47±0.09   |   70.45±0.25   |   61.32±0.33   |   84.15±0.20   |   79.05±0.12   |   92.90±0.05   |   89.25±0.04   |   87.06±0.07   |   80.12±0.05   |
> | NE+LastN      |   92.36±0.18   |   87.92±0.27   |   92.37±0.08   |   87.25±0.12   |   63.23±0.75   |   54.15±0.92   |   79.15±0.16   |   71.71±0.28   |   90.62±0.12   |   86.12±0.05   |   85.02±0.06   |   77.47±0.08   |
> | NE+RandomSN   |   92.65±0.21   |   88.10±0.15   |   92.15±0.23   |   88.76±0.26   |   64.39±1.08   |   54.88±1.35   |   81.39±0.65   |   73.15±0.81   |   91.08±0.30   |   86.57±0.19   |   84.77±0.24   |   77.96±0.16   |
> | **MIN**       | **95.12±0.05** | **92.12±0.16** | **94.00±0.15** | **91.22±0.18** | **72.89±0.45** | **64.32±0.89** | **85.18±0.19** | **79.75±0.07** | **93.36±0.07** | **90.04±0.06** | **87.36±0.04** | **80.55±0.10** |
>
> ## Reply to W3.2
>
> >W3.2: It would be insightful to analyze or visualize the final learned weights $w$ after incremental tasks are completed.
>
> **Reply**: Thanks for your valuable advice. Due to the limitation of rebuttal rules, we can not upload any figures. We intend to add the visualizations of the learned weights $w$ across 10 tasks in the format of the heat map to illustrate the dynamics of weight $w$.

---

> > ### Comment · Reviewer_MbJ6 · 2025-08-06
> >
> > Thanks to the authors' efforts to address my concern.
> > I think most of my major concerns are successfully addressed.
> >
> > However, I still think that changing $z$ to $z+\epsilon$ effectively modifies the function $\mathcal{F}()$ across different tasks. At present, the manuscript does not include a derivation that accounts for this inter-task variation in $\mathcal{F}()$. This makes it difficult to ascertain if the relationship described in Eq. 3 remains valid.
> >
> > Overall, I will raise my score to `borderline accept' because some of my concerns are addressed, and I encourage the author to give rigorous proof of Eq. 3.

---

> > > ### Author Response · Authors · 2025-08-06
> > >
> > > Thanks sincerly for your positive feedback. Converting $z$ to $z+\epsilon$ is just an explanation for the effectness of this mechanism. Appreciate again for your constructive comments. We will try our best to add more derivations and explanations for Eq. (3) to provide a rigorous proof.

---

> ### Author Response · Authors · 2025-08-05
>
> Dear Reviewer,
>
> Thank you sincerely for your previous comments.
>
> I wanted to follow up to ensure that we have adequately addressed all of your concerns from the initial review. Could you please let us know if our explanations and revisions have satisfactorily resolved the issues you raised? If there are any remaining concerns or if you need further clarification on any points, we would be happy to provide additional information.
>
> We appreciate your time and valuable feedback throughout this review process.
>
> Best regards,
>
> All of Authors.

---

### Comment · Area_Chair_TbQV · 2025-08-04
**Reminder: Author–Reviewer Discussion**

Dear reviewers,

A friendly reminder that the author–reviewer discussion period will close at August 6, 11:59 pm, AoE. The current mixed ratings on this submission make your final justification particularly valuable. Please engage with the authors’ questions and comments and update your Final Justification accordingly.

Thank you for your time and engagement.

Best regards,

AC

---

### Decision · Program_Chairs · 2025-09-17

**Decision:**

Accept (poster)

**Comment:**

This paper proposes a simple yet effective method for class-incremental learning (CIL) with pre-trained models by introducing a mixture of noise strategies during finetuning. Specifically, it injects different types of noise (Gaussian, dropout, etc.) into various components of the model (features, outputs, logits) to improve robustness and mitigate overfitting to new tasks. The authors show strong empirical results across standard benchmarks and model backbones, demonstrating both improved accuracy and reduced forgetting. The method is lightweight, compatible with existing models, and easy to implement.

Reviewers generally appreciated the empirical gains and simplicity of the approach. However, several concerns were raised about novelty (i.e., noise injection is not new) and the lack of deeper theoretical analysis or broader contextualization within the literature. While the authors provided useful clarifications and additional analysis in the rebuttal (e.g., on ablations and comparison to related methods), some concerns about motivation and positioning remain. Overall, due to its empirical effectiveness, broad applicability, and methodological clarity, I recommend acceptance, though not at the level of spotlight.